# FEDERATED ADAPTER ON FOUNDATION MODELS: AN OUT-OF-DISTRIBUTION APPROACH

## ABSTRACT

As foundation models gain increasing attention from both academic and industrial communities, Federated Foundation Models (FedFM) have emerged as a privacy-preserving approach for collaboratively fine-tuning models in federated learning (FL) frameworks using distributed datasets across multiple clients. A key challenge for FedFM, given the versatile nature of foundation models, is addressing out-of-distribution (OOD) generalization, where unseen tasks or clients may exhibit distribution shifts leading to suboptimal performance. Although numerous studies have explored OOD generalization in conventional FL, these methods are inadequate for FedFM due to the challenges posed by large parameter scales and increased data heterogeneity, where large parameter scales would result in high computational and communication costs while increased data heterogeneity like cross-domain would lead to suboptimal performance of the aggregated global model on individual client distributions. To bridge this gap, we propose a new method, called FedOA, to enhance the OOD generalization of FedFM under these conditions. Specifically, our method employs adapter-based parameter-efficient fine-tuning methods for efficient learning, and introduces an additional personalized model with a feature distance-based regularization to ensure distribution alignment and provide OOD generalization guarantees for each client. Theoretically, we demonstrate that the conventional aggregated global model in FedFM inherently retains OOD generalization capabilities, and our proposed method enhances the personalized model's OOD generalization through regularization informed by the global model, with proven convergence under general non-convex settings. Empirically, the effectiveness of the proposed method is validated on benchmark datasets across various NLP tasks.

## 1 INTRODUCTION

Recently, foundation models have garnered considerable attention from both academic and industrial communities due to their versatile capabilities in handling a wide range of downstream tasks. Despite their advantages, these models predominantly rely on large volumes of publicly available data, which poses significant challenges related to the exhaustion of public data resources. To mitigate these issues, Federated Foundation Models (FedFM) (Zhuang et al., 2023; Yu et al., 2023) have been proposed as a promising solution. By leveraging the federated learning (FL) framework, FedFM facilitates the distributed training of foundation models across multiple devices or data sources, ensuring that private data remains localized without being directly shared.

Out-of-distribution (OOD) generalization constitutes a pivotal research challenge that endeavors to train models capable of performing robustly on data exhibiting distributions difference from those seen during training. This challenge has been extensively explored across various centralized research areas (Liu et al., 2021b; Arjovsky, 2020), and recent scholarly efforts have extended these methodologies to federated learning frameworks (Li et al., 2023a; Yuan et al., 2021), in which some unseen (non-participation during training) tasks/clients may exhibit distribution shifts leading to suboptimal performance of the conventional FL methods. One prevalent approach to address this issue involves adapting invariant learning (Arjovsky et al., 2019; Koyama & Yamaguchi, 2020) in FL to identify and learn invariant features that are consistent across all distributions. For example, in a model trained to classify cows and camels, invariant learning encourages the model to focus on invariant features, such as animal shapes, rather than spurious correlations, such as backgrounds ( as-

sociating green landscapes with cows), which enhances the model's ability to generalize effectively to new environments (such as cows on a sandy beach).

Although these invariant learning approaches for addressing OOD generalization in conventional FL are promising, they may not be optimal for federated foundation models. A key distinction between FedFM and conventional FL lies in the scale of parameters involved (Ren et al., 2024). Unlike conventional FL, which primarily focuses on smaller models, FedFM typically utilizes foundation models with billions of parameters. This scale can result in substantial communication and computation costs when attempting to operate directly on the entire model. Another significant challenge for FedFM is the exploration of more heterogeneous data, such as cross-domain data, due to the versatile nature of foundation models, which are designed to handle a variety of downstream tasks in real-world applications (Liu et al., 2023). Given the vast parameter count and the increased data heterogeneity in FedFM, it is crucial to explore innovative approaches that can effectively address OOD generalization in FedFM while minimizing computational and communication overhead in the increased data heterogeneity scenarios.

Previous work (Du et al., 2024) has provided an initial analysis of the OOD generalization capability of federated foundation models through a series of robustness analysis experiments and introduced a general noisy projection-based robust aggregation algorithm. However, this approach remains rooted in the general non-IID (heterogeneous label distributions) setting typical of conventional FL and lacks a comprehensive theoretical analysis. To address these limitations, we propose FedOA, a novel framework that adapts invariant learning for OOD generalization in FedFM while addressing the substantial communication and computation costs associated with more heterogeneous scenarios. Our approach begins by revisiting existing invariant learning techniques in conventional FL, reformulating them into a unified optimization framework. We then theoretically analyze the generalization bounds of both the conventional aggregated global model and the personalized model in FedFM, demonstrating that the conventional aggregated global model in FedFM inherently retains OOD generalization ability. This motivates our approach to enhance the OOD generalization of the personalized model in FedFM by leveraging the global model. Specifically, we employ adapter-based parameter-efficient fine-tuning (PEFT) methods (Hu et al., 2023) to facilitate efficient learning by tuning and communicating only a small subset of the model parameters. Given the increased heterogeneity and significant distribution shifts across clients in FedFM, we further incorporate personalized models to better address individual client needs and introduce a feature distance-based regularization term to enhance OOD generalization and further address large parameter scales. Finally, we establish a new theoretical framework to analyze the convergence of our method in FedFM. Our contributions are summarized below.

- We introduce a new method, namely FedOA, to learn invariant features for addressing the OOD generalization of FedFM with large parameter scales in increased data heterogeneity scenarios.

- We theoretically demonstrate that the conventional aggregated global model in FedFM inherently retains OOD generalization ability, and FedOA is expected to enhance OOD generalization through feature distance-based regularization. We also present the convergence results for FedOA under general non-convex settings.

- We conduct an experimental analysis using heterogeneous FedFM benchmarks across diverse NLP tasks. Empirical outcomes reveal that our method attains state-of-the-art performance, underscoring its superior OOD generalization capabilities than existing methods.

## 2 PRELIMINARIES AND CHALLENGES

### 2.1 PRELIMINARIES

Let $\mathcal{X}$ denote the feature space and $\mathcal{Y}$ the label space. There are often families of probability distributions $\{P_e\}_{e \in \mathcal{E}}$ over the space $\mathcal{X} \times \mathcal{Y}$, where the indices $e \in \mathcal{E}$ represent different environments (also referred as "domains"). Each distribution $P_e$ can be denoted as $(X^e, Y^e) \sim P_e$. $\mathcal{E}_{all}$ is the collection of all possible environments, with $\mathcal{E}_{train}, \mathcal{E}_{test} \subseteq \mathcal{E}_{all}$ as training and testing environments respectively. The notations related to OOD generalization are delineated in the first part of Table 1, whereas the latter part elucidates components relevant to federated learning.

Table 1: Table of partial notations.

| Components | Notation | Definition |
|---|---|---|
| OOD | $(X, Y)$ | Random variables of inputs and outputs |
| | $f_\theta$ | Hypothesis with parameter $\theta$ |
| | $\ell(f(X), Y)$ | Loss function |
| | $(X^e, Y^e) \sim P_e$ | Probability distribution of environment $e$ |
| | $\mathcal{E}$ | Collection of environments $e$ |
| | $\mathcal{R}(f) = \mathbb{E}_{(X,Y) \sim P}[\ell(f(X), Y)]$ | Expected risk of model $f$ |
| FL | $S_e, |S_e|$ | The dataset and its size on Client $e$ |
| | $\xi \sim S$ | Batch of samples from dataset $S$ |
| | $K$ | Number of local update steps |
| | $T$ | Number of communication rounds |
| | $\eta_l, \eta_g$ | Local and global learning rates |
| | $R(f) = \frac{1}{|S|} \sum_{(x_i, y_i) \in S} \ell(f(x_i), y_i)$ | Empirical risk of model $f$ over data $S$ |

**The Objective of OOD Generalization.** In practical settings, there is often such a case in which test data originate from distributions that differ from those of the training data. OOD generalization is a research domain that specifically addresses these discrepancies. Following the conventional methodologies (Arjovsky, 2020), we assume that the distribution of the test data belongs to $\mathcal{E}_{all}$ and the objective of OOD generalization is to minimize the worst case over all potential test distributions, which can be formulated as:

$$\min_f \max_{e \in \mathcal{E}_{all}} \mathcal{R}_e(f) \qquad (1)$$

where $\mathcal{R}_e(f) = \mathbb{E}_{(X^e, Y^e) \sim P_e}[\ell(f(X^e), Y^e)]$, $f$ is the model and $\ell$ is the loss function.

**OOD Generalization in FL.** In FL, the task in each client can be taken as an environment $e$, which holds a local dataset $S_e$ driven from the distribution $P_e$. Consequently, tasks in training clients can be taken as the collection of $\mathcal{E}_{train}$, and $\mathcal{E}_{all}$ represents all possible tasks/clients. The objective of OOD generalization in FL, therefore, aligns with the general objective outlined in equation (1). Specifically, due to the distributed nature of FL, out-of-distribution scenarios can occur within individual clients (**intra-client**) or across different clients (**inter-client**) (Yuan et al., 2021). Intra-client OOD scenarios refer to distribution shifts that occur in unseen tasks within the same client, whereas inter-client OOD scenarios refer to distribution shifts that arise in previously unseen clients.

Given the long-standing focus on representation learning in machine learning, existing work on OOD generalization in FL primarily concentrates on adopting invariant learning (Arjovsky et al., 2019; Koyama & Yamaguchi, 2020; Liu et al., 2021a), which seeks to learn features that remain consistent across all environments. In the context of representation learning, the model architecture is typically divided into two distinct components: a feature encoder $\Phi$ to learn representations and a head $\boldsymbol{w}$ to get the final predictive outcomes. This can be mathematically represented as $f_\theta = \boldsymbol{w}_w \circ \Phi_\phi$, where $\theta = (w, \phi)$. These invariant learning methods operate under the assumption that the representations extracted by the encoder are invariant across all different environments, which can be formalized in the following manner:

**Assumption 1.** *There exists a representation $\Phi$ such that for all $e, e' \in \mathcal{E}_{all}$ and all $\boldsymbol{z}$ in the intersection of the supports $Supp(P(\Phi(X^e))) \cap Supp(P(\Phi(X^{e'})))$, we have*

$$\mathbb{E}[Y^e | \Phi(X^e) = \boldsymbol{z}] = \mathbb{E}[Y^{e'} | \Phi(X^{e'}) = \boldsymbol{z}].$$

Under this assumption, the feature encoder is tasked with managing the heterogeneity among different environments (clients) to learn invariant features. Consequently, the integration of invariant learning within FL frameworks can be uniformly expressed as follows:

$$\min_\Phi \sum_{e \in \mathcal{E}_{train}} \alpha_e R_e(\Phi) \qquad (2)$$

where $\alpha_e$ denotes the importance weight for the environment (client) $e$ and $R_e(\Phi)$ denotes the empirical risk of $\Phi$ over $S_e$. Specially, unlike the empirical risk of the overall model $f$ computing

the loss between predicted logits and actual labels $y$, the empirical risk of $\Phi$ calculates using similar or consistent features $z$ (invariant features) as labels, focusing on the feature space. Based on this framework, various methods have been proposed. For instance, some works (Guo et al., 2023; Tang et al., 2023) employ the objective (2) using a similar or identical head, while others (Zhang et al., 2021; Tan et al., 2024) focus on adversarial/contrastive learning to directly optimize the feature encoder. Additionally, other studies (Deng et al., 2020; Zhang et al., 2023b) explore different importance weight strategies to learn more robust features.

## 2.2 CHALLENGES OF OOD GENERALIZATION IN FEDFM

Federated foundation models represent an emerging research area that introduces new challenges beyond those encountered in conventional FL. **(1) Large Parameters:** conventional FL typically focuses on smaller models, such as convolutional neural networks (CNNs), which involve relatively few parameters (e.g., ResNet (He et al., 2016) with approximately 25 million parameters). In contrast, FedFM deals with foundation models with parameter counts that can reach into the billions; for instance, models like LLAMA (Touvron et al., 2023) contain over 7 billion parameters. The significantly larger parameter scale in FedFM introduces substantial challenges in terms of computation and communication costs during training. As a result, the methods traditionally used in conventional FL are suboptimal for FedFM, necessitating the development of more parameter-efficient learning approaches. **(2) Increased Data Heterogeneity.** Foundation models are designed to address a wide range of downstream tasks, leading FedFM to encounter more heterogeneous data than conventional FL (Zhuang et al., 2023; Yu et al., 2023; Ren et al., 2024; Charles et al., 2024). Unlike conventional FL, which typically deals with label or feature distribution heterogeneity across clients, FedFM would have to manage increased data heterogeneity stemming from cross-dataset or cross-task distribution heterogeneity, collectively referred to as cross-domain distribution heterogeneity. Given this increased data heterogeneity, there is a critical need for personalized models that can effectively adapt to the diverse distributions across different clients, thereby enhancing overall performance. However, existing methods for personalization in conventional FL often fall short in terms of generalization (Jiang & Lin, 2023; Xie et al., 2024), making them less effective for the versatile applications required in FedFM. This underscores the need for the development of personalized federated foundation models that can achieve better generalization in scenarios characterized by increased data heterogeneity.

As analyzed above, due to the challenges posed by large parameters and increased data heterogeneity, traditional methods for addressing OOD generalization in conventional FL are inadequate for direct application in FedFM. *This motivates the development of an efficient adapter-based personalized FedFM method with OOD generalization guarantees.*

## 3 METHOD

To address large parameter scale and increased data heterogeneity challenges in FedFM, we propose an adapter-based personalized FedFM method with OOD generalization guarantees. In this section, we starts by analyzing the generalization bounds of both the conventional global and personalized models in FedFM, then outline the optimization objective of our method that facilitates the learning of invariant features through feature distance-based regularization and the detailed algorithm, finally discuss our method's deployment in both intra-client and inter-client OOD scenarios.

### 3.1 GENERALIZATION ANALYSIS

We begin by analyzing the generalization bound of the conventional aggregated global model in FL. The aggregated global hypothesis $f_g$ is defined with the objective $f_g = \arg\min_{f \in \mathcal{F}} \sum_{e \in \mathcal{E}_{train}} \alpha_e R_e(f)$. Following previous work (Konstantinov & Lampert, 2019), for any testing environment $e' \in \mathcal{E}_{all}$, the generalization bound of the global hypothesis $f_g$ is primarily constrained by the discrepancy $\sum_{e \in \mathcal{E}_{train}} \alpha_e d_{\mathcal{F}}(P_e, P_{e'})$, where $d_{\mathcal{F}}(P_e, P_{e'}) = Supp_{f \in \mathcal{F}}(|\mathcal{R}_e(f) - \mathcal{R}_{e'}(f)|)$.

**Theorem 1.** (Conventional aggregated global model in FedFM inherently retains OOD generalization ability). *In FedFM, we consider learning the global hypothesis $f_g = (\boldsymbol{w}, \Phi_g)$. Since foundation models are pre-trained with massive data in one unified format, this results in an optimal and*

*fixed head $\boldsymbol{w}$ towards all tasks during tuning (Hu et al., 2023), that is, $\boldsymbol{w} \in \arg\min_{\boldsymbol{w}} R_e(\boldsymbol{w}, \Phi_g)$ for all $e \in \mathcal{E}_{all}$. Accordingly, the objective of $f_g$ can be further formulated as objective (2) to learn invariant representations $\boldsymbol{z} = \Phi_g(X)$. Therefore, the discrepancy $d_{\mathcal{F}}(P_e, P_{e'}) = Supp_{f \in \mathcal{F}}(|\mathbb{E}[\ell(\boldsymbol{w}(\boldsymbol{z})), Y^e] - \mathbb{E}[\ell(\boldsymbol{w}(\boldsymbol{z})), Y^{e'}]|)$ approaches zero if $\boldsymbol{z}$ is an invariant representation according to Assumption 1.*

Due to the increased data heterogeneity in FedFM, personalized models are essential to align with the specific distribution of each client for individual user preferences. To address this, we further analyze the generalization bound of the conventional personalized model in FedFM. As the head $\boldsymbol{w}$ remains fixed during the turning of foundation models, the difference between personalized hypothesis $f_e = (\boldsymbol{w}, \Phi_e)$ and global hypothesis $f_g = (\boldsymbol{w}, \Phi_g)$ lies in the feature encoder $\Phi$.

**Theorem 2.** (Generalization bound of the personalized model in FedFM is further constrained by the invariant feature distance.) *In FedFM, we consider learning the personalized hypothesis $f_e = (\boldsymbol{w}, \Phi_e)$. Given that the generalization bound for the global hypothesis $f_g$ has been established in previous work (Konstantinov & Lampert, 2019), we primarily need to examine the distance $|\mathcal{R}_{e'}(f_e) - \mathcal{R}_{e'}(f_g)| = |\mathbb{E}[\ell(\boldsymbol{w}(\Phi_e(X^{e'}))), Y^{e'}] - \mathbb{E}[\ell(\boldsymbol{w}(\Phi_g(X^{e'}))), Y^{e'}]|$ to determine the generalization bound for the personalized hypothesis $f_e$. Therefore, based on Assumption 1, the generalization bound of the personalized model in FedFM is further constrained by $\mathbb{E}[D(\Phi_e(X^{e'}), \Phi_g(X^{e'}))]$, where $D$ denotes the feature distance function.*

As shown in Theorem 2, the generalization bound of the conventional personalized model in FedFM is further constrained by the feature distance $\mathbb{E}[D(\Phi_e(X^{e'}), \Phi_g(X^{e'}))]$. Since it is challenging to directly quantify this distance, we are motivated to optimize it during the learning process of the personalized model in FedFM to achieve a tighter generalization bound. However, due to the inaccessibility of unseen environments' data during training, we instead optimize the feature distance using the available training environments and incorporate this distance as a regularization term in the learning of the personalized model. Given that the aggregated global model captures invariant features across all environments, aligning the personalized model's features through this regularization term implicitly encourages the personalized model to align with the global model for invariant feature learning, thereby enhancing its OOD generalization ability. For more detailed proofs of the generalization bound, please refer to Appendix D.

## 3.2 OPTIMIZATION OBJECTIVE

As discussed in Section 2.2, the large parameter scale and increased data heterogeneity present two key challenges for FedFM. To address the issue of increased data heterogeneity, we introduce an additional personalized model for each client, tailored to align with specific data distributions, thereby enhancing overall performance. Simultaneously, to ensure the versatility of foundation models, we incorporate a feature distance-based regularization term inspired by the generalization analysis in Section 3.1. This regularization leverages insights from the aggregated global model to enhance the OOD generalization capability of the personalized model. In addition, to mitigate the challenges posed by large parameter scale in FedFM, we employ adapter-based PEFT methods (Hu et al., 2023). These PEFT methods strategically divide the parameters $\theta$ of foundation models into two parts: the majority frozen part $\theta_f$ and a small tunable part $\Delta\theta$, represented as $\theta = (\theta_f, \Delta\theta)$. For example, in employing the LoRA Hu et al. (2021), low-rank matrices are integrated to decompose parameters into frozen and trainable parts as $\theta = \theta_f + \Delta\theta = \theta_f + \Delta\theta^A \Delta\theta^B$. During the learning phase in FedFM with PEFT methods, only the small part $\Delta\theta$ is updated and communicated across the federated network to reduce the communication overhead and computational burden.

**Objective.** We focus exclusively on the feature encoder $\Phi$, which consists of tunable adapter $\phi$ and other frozen parts $\phi_{frozen}$, disregarding the fixed head $\boldsymbol{w}$. FedOA is designed to learn a personalized $\Phi_e$ for each client, characterized by a unique dataset denoted as $S_e$, while ensuring OOD generalization from the aggregation $\Phi_g$ with regularization,

$$\min_{\Phi_e} \quad R_e(\Phi_e) + \lambda D(\Phi_e(X^e), \Phi_g^*(X^e))$$

$$s.t. \quad \Phi_g^* \in \arg\min_{\Phi} \sum_{e \in \mathcal{E}_{train}} \alpha_e R_e(\Phi_g) \tag{3}$$

where $D$ represents a function to measure distance and $\lambda$ controls the interpolation between personalized and global models.

**Why feature distance-based regularization?** In conventional FL, parameter regularization is the most preventive method (Li et al., 2020; 2021a; T Dinh et al., 2020; Xie et al., 2024). However, for FedFM, parameter regularization would lead to high computation costs and unintended results. Firstly, due to the large scale of parameters in FedFM, applying regularization directly to all parameters incurs substantial computational costs. In contrast, feature vectors are much smaller in size compared to the full parameter set of an FedFM, making feature distance-based regularization more storage- and computation-efficient in this context. Secondly, while parameter regularization could be applied between adapters to reduce computational overhead, it often leads to unintended results due to the varying structures and combinations of PEFT methods used in FedFM as shown in previous work (Sun et al., 2024). For instance, the LoRA method in PEFT involves two low-rank matrices that are combined multiplicatively; regularizing each matrix separately diverges from the objective of jointly optimizing them. In contrast, feature distance-based regularization avoids this discordance, as it implicitly guides the learning of adapter parameters without directly manipulating the adapters themselves. Additionally, unlike previous methods Zhou et al. (2023) that utilize prototypes for regularization requiring a finite categorization, feature distance-based regularization are not bound by a set number of categories and learn invariant features autonomously across different environments by the feature encoder, which is more suitable for federated foundation models in OOD scenarios due to open-vocabulary tasks inherently (e.g. the categories of real-world images are effectively infinite).

## 3.3 ALGORITHM

As outlined in algorithm 1, our method optimizes the personalized adapter and the aggregated global adapter iteratively for each round. **On the server side**, for each communication round $t \in [T]$, a subset of clients $\mathcal{E}_t$ is selected. In the first round $t = 0$, the server initializes the global adapter $\Phi_g$ with parameters $\phi_g^0$ and broadcasts the initialized global adapter to the selected clients. In subsequent communication rounds $t \in \{1, .., T-1\}$, after receiving the returned global adapter $\phi_g^{t-1,e}$ from each selected client, the server aggregates these adapters across all selected clients to obtain the updated global adapter for the next round, denoted as $\phi_g^t = \sum_{e \in \mathcal{E}_t} \alpha_e \phi_g^{t-1,e}$. **On the client side**, each client maintains two adapters: a personalized adapter $\Phi_e$ with parameters $\phi_e$ and a global adapter $\Phi_g^e$ with parameters $\phi_g^e$. For each communication round $t \in [T]$, the client initializes the personalized adapter as $\phi_{e,0}^t = \phi_e^{t-1}$ and performs $K$ local update steps to obtain $\phi_e^t = \phi_{e,K}^t$. Similarly, the global adapter in each client is initiated as $\phi_g^e = \phi_g^{t-1}$ to obtain $\phi_g^{t-1,e}$. Specifically, the updated global adapter $\phi_g^{t-1,e}$ is sent back to the server for aggregation, while the personalized adapter $\phi_e^t$ remains local without communication.

**Remark.** Our framework is flexible and can be adapted to any aggregation algorithm, any adapter-based PEFT method, and any transformer-based foundation model by simply substituting the corresponding components. In this paper, we utilize FedAVG (McMahan et al., 2017), LoRA (Hu et al., 2021), and large language models (LLMs) (Zhao et al., 2023) as illustrative examples to demonstrate the framework of our method.

## 3.4 INFERENCE

As highlighted in previous work (Yuan et al., 2021), OOD scenarios can occur either within the same client (**intra-client**) or across different clients (**inter-client**). In intra-client OOD scenarios, the test data exhibits distribution shifts from the training data in the same client, while in inter-client OOD scenarios, new clients' data experience distribution shifts from these training clients. Our proposed method is capable of addressing both types of OOD scenarios. For intra-client OOD scenarios, the learned personalized model can be directly deployed to handle the distribution shifts within the same client. For inter-client OOD scenarios, the aggregated global model can be deployed to manage distribution shifts among different clients. As analyzed in Section 3.1, conventional aggregation in FedFM is inherently capable of achieving OOD generalization, while conventional personalized adaptation methods often lack this generalization guarantee, resulting in suboptimal performance

---

**Algorithm 1** FedOA

---

**Input**: Clients $\mathcal{E}_{train}$, local datasets $\{S_e\}_{e \in \mathcal{E}_{train}}$, communication rounds $T$, local update steps $K$
**Output**: Personalized adapters $\{\phi_e\}_{e \in \mathcal{E}_{train}}$ and global adapter $\phi_g$

1: **for** $t = 0, ..., T - 1$ **do**
2:     Server randomly selects a subset of devices $\mathcal{E}_t$, and sends $\phi_g^{t-1}$ to them
3:     **for** client $e \in \mathcal{E}_t$ in parallel **do**
4:         **for** $k = 0, ..., K - 1$ **do**
5:             Sample mini-batch $\xi$ from local data $S_e$
6:             // update personalized adapter
7:             $\phi_{e,k}^t = \phi_{e,k-1}^t - \eta_l \nabla(R_e(\phi_{e,k-1}^t; \xi) + \lambda D(\Phi(\phi_{e,k-1}^t; \xi), \Phi(\phi_g^{t-1}; \xi)))$
8:         **end for**
9:         // update global adapter
10:        $\phi_g^{t-1,e} = \phi_g^{t-1} - \eta_g \nabla R_e(\phi_g^{t-1})$
11:        Send $\phi_g^{t-1,e}$ back to server
12:     **end for**
13:     Server aggregates $\phi_g^t = \sum_{e \in \mathcal{E}_t} \alpha_e \phi_g^{t-1,e}$
14: **end for**

---

in intra-client OOD scenarios. Therefore, our experiment primarily focuses on intra-client OOD scenarios to evaluate the effectiveness of the proposed personalized adaptation approach in handling these distribution shifts.

# 4 CONVERGENCE ANALYSIS

In this section, we delve into the convergence analysis of the proposed method. For the purpose of clarity in our analysis, we restrict our focus to the small tunable part of parameters $\phi$, while excluding other parameters that remain frozen. We first state several standard assumptions on the function.

**Assumption 2.** (Smoothness). *For all clients $e$, we assume that $R_e(\phi)$ and $\Phi_e$ are $L$-Lipschitz smoothness, as follows when $\forall \phi, \phi'$:*

$$
\begin{aligned}
||\nabla R_e(\phi) - \nabla R_e(\phi')|| &\leq L||\phi - \phi'||, \\
||\nabla \Phi_e(\phi) - \nabla \Phi_e(\phi')|| &\leq L||\phi - \phi'||.
\end{aligned} \tag{4}
$$

**Assumption 3.** (Unbiased gradient estimator and Bounded gradients). *For all clients $e$, we assume that the expectation of stochastic gradient $\nabla R_e(\phi; \xi)$ and $\nabla \Phi_e(\phi; \xi)$ are unbiased estimators of the local gradients $\nabla R_e(\phi)$ and $\nabla \Phi_e(\phi)$, and are uniformly bounded by $\sigma^2$. For $\forall \phi$, we have*

$$
\begin{aligned}
\mathbb{E}||\nabla R_e(\phi; \xi)|| &= \nabla R_e(\phi), \mathbb{E}||\nabla \Phi_e(\phi; \xi)|| = \nabla \Phi_e(\phi); \\
\mathbb{E}||\nabla R_e(\phi; \xi)||^2 &\leq \sigma^2, \mathbb{E}||\nabla \Phi_e(\phi; \xi)||^2 \leq \sigma^2.
\end{aligned} \tag{5}
$$

**Assumption 4.** (Bounded Diversity). *For all clients $e$, we assume that the variance of the local gradient to the global gradient is bounded by $G$. For $\forall e, \phi$, we have*

$$
||\nabla R_e(\phi) - \nabla R(\phi)|| \leq G. \tag{6}
$$

Assumption 2 delineates the smoothness of the local risk function, a technique well-established in the optimization analysis (Crane & Roosta, 2019; Elgabli et al., 2022). Given the dependence of our method on the representation function, we also assume the representation function $\Phi$ is $L$-smoothness. Assumption 3 establishes a boundary on the variance of the stochastic gradient, an approach commonly used in stochastic optimization analysis (Karimireddy et al., 2021; Wang et al., 2021). Similarly, we also bound the stochastic gradient of the representation function $\Phi$ in our analysis. Assumption 4 bounds the variance of local gradients relative to the global gradient, a method extensively utilized to quantify statistical heterogeneity in the federated learning (Fallah et al., 2020).

For the convenience of analysis, we use L2-distance as the distance function $D$ of the regularization term in equation (3). We now present the convergence results of FedOA for the general non-convex case.

**Theorem 3.** *Suppose that Assumption 2, 3 and 4 hold true, our method updates with constant local and global step-size such that $\eta_l \leq \frac{1}{8\sqrt{3(1+3T)T(1+2K)K\lambda\sigma L}}$ and $\eta_g \leq \frac{1}{2\sqrt{6(1+3T)TL}}$. Then, the sequence of iterates generated by our method satisfies:*

$$\frac{1}{T}\sum_{t=1}^{T}\mathbb{E}||\nabla R_e(\phi_e^{t-1})||^2 \leq \frac{2(\mathbb{E}R_e(\phi_e^0) - \mathbb{E}R_e(\phi_e^*))}{T} + 8K(1+2K)(L-1)(1+12\lambda^2 L^2 M^2)\sigma^2\eta_l^2$$

$$+ 256K(1+2K)T(1+3T)\lambda^2\sigma^2(L-1)L^2G^2\eta_l^2\eta_g^2$$

(7)

*If we choose the step sizes $\eta_l = \mathcal{O}(\frac{1}{TKL\sigma})$ and $\eta_g = \mathcal{O}(\frac{1}{TL})$, we have the convergence rates of our method as follows*

$$\frac{1}{T}\sum_{t=1}^{T}\mathbb{E}||\nabla R_e(\phi_e^{t-1})||^2 = \mathcal{O}(\frac{(\mathbb{E}R_e(\phi_e^0) - \mathbb{E}R_e(\phi_e^*))}{T}, \frac{1+\lambda^2 L^2 M^2}{T^2 L}, \frac{\lambda^2 G^2}{T^2 L})$$

(8)

As analyzed above, FedOA converges to a stationary point at a rate of $\mathcal{O}(\frac{1}{T})$. The heterogeneity between clients and between the personalized and global models is captured by $G$ and $M$, respectively. The impact of these heterogeneities can be reduced by increasing $T$. Similarly, the interpolation between the personalized and global models, controlled by $\lambda$, also becomes less significant as $T$ increases. The full proof of these results is provided in Appendix E.

## 5 EXPERIMENTS

In this section, we present experiments to evaluate the performance of our proposed FedOA method and answer the following questions. **Q1:** Can the conventional aggregated global model in FedFM demonstrate superior OOD generalization ability compared to the centralized model? **Q2:** In increased heterogeneity scenarios, can FedOA achieve improved OOD generalization performance relative to existing generalization methods in conventional FL?

### 5.1 EXPERIMENT SETTING

**Datasets.** We construct four federated datasets, each centered around a distinct task, derived from the Flan (Wei et al., 2021), which encompasses a wide range of NLP tasks from over 60 datasets designed for instruction tuning. The tasks selected include Entailment, Sentiment, Paraphrase and Reading Comprehension, each of which consists of two distinct datasets from different domains, reflecting the increased heterogeneity characteristic of FedFM. *Since foundation models standardize all tasks into a uniform format, we can treat all tasks as a single unified task, with the original distinct tasks viewed as different distributions of this unified task.* Therefore, to better align with OOD settings, we perform the "leave-one-task-out" strategy, where one task is set aside as the test environment, while the remaining are used as training environments. ROGUE-1 is used as the evaluation metric and more details are in Appendix C.1.

**Baselines and Implementation.** We compare our methods with the following baselines based on the same model architecture: 1) global models: centralized model and FedIT (Zhang et al., 2023a); 2) personalized models: pFedMe (T Dinh et al., 2020) and FedLoRA (Yi et al., 2023); 3) personalized models with generalization guarantees: PERADA (Xie et al., 2024) and FedSDR (Tang et al., 2023). The centralized model is trained on all data of training environments in one center. Here, we adapt the training paradigm in pFedMe, FedLoRA, PERADA and FedSDR to federated foundation models with NLP tasks. We distribute data between clients based on the dataset for data heterogeneity, with the number of training clients as $|\mathcal{E}_{train}| = 6$. To better evaluate the effectiveness of methods, we assume that all clients are activated for every communication round and set the communication round $T = 20$. The alpaca-LoRA[1] is adapted as the base model initialized with LLaMA-7B[2]. We set $\lambda = 0.5$ and choose L2-distance as the distance function $D$. More details about baselines are in Appendix C.2.

---

[1]https://github.com/tloen/alpaca-lora
[2]https://huggingface.co/huggyllama/llama-7b

Table 2: OOD results of different models using "leave-one-task-out" validation. Centralized and FedIT are tested on a single global model, while the remaining models are tested on personalized models with average results reported. Reading Com represents the Reading Comprehension task.

| Methods | Entailment | Sentiment | Paraphrase | Reading Com | Average |
|---|---|---|---|---|---|
| Centralized | 42.00 | 76.75 | 43.25 | 64.16 | 56.54 |
| FedIT | **44.00** | 80.00 | 43.00 | 65.72 | 58.18 |
| pFedMe | 36.60 | 76.13 | 44.21 | 50.91 | 51.96 |
| FedLoRA | 40.13 | 78.29 | 44.17 | 63.40 | 56.50 |
| PRADA | 36.52 | 76.94 | 44.22 | 53.98 | 52.92 |
| FedSDR | 37.05 | 66.15 | 43.26 | 43.08 | 47.39 |
| FedOA | 40.62 | **82.21** | **45.46** | **67.61** | **58.97** |

## 5.2 MAIN RESULTS

**Conventional aggregated global model in FedFM achieves better OOD generalization performance than that in centralized setting.** In response to **Q1**, we compare the OOD generalization performance of the global model in FedFM with that in a centralized setting on four datasets. Specifically, we take FedIT as a baseline method in FedFM for learning the aggregated global model, which adapts FedAVG with the PEFT method LoRA for instruction learning. In this experiment, our proposed FedOA follows the same global model learning process as FedIT, while FedOA is designed to be adaptable to any other global model learning algorithms as well. As shown in Table 2, FedIT exhibits superior performance in OOD generalization compared to the centralized model, indicating that conventional aggregation in FedFM can indeed achieve a degree of OOD generalization. This finding is consistent with the theoretical analysis presented in Theorem 1.

**FedOA demonstrates better OOD generalization performance compared to other baselines.** In response to **Q2**, we compare FedOA with different baselines to evaluate the OOD generalization performance on four datasets. Compared to personalized models, as shown in Table 2, FedOA stands out as the most effective among all personalized models, which suggests that incorporating feature distance-based regularization from the global adapter is crucial for invariant feature learning to improve OOD generalization performance. Additionally, FedLoRA ranks second, as its further tuning of the learned global model introduces minimal updates, thus maintaining certain OOD generalization ability from the global model. The underperformance of PERADA and pFedMe, which rely on parameter regularization, indicates that this regularization is unsuitable for FedFM due to the discordance between regularization operation and optimization objective. Moreover, the recent benchmark FedSDR for OOD generalization in conventional FL performs poorly, highlighting the inadequacy of conventional FL methods in handling FedFM's increased heterogeneity. Compared to global models, FedOA leverages the global model's OOD generalization ability to guide personalized models, resulting in slightly better average OOD generalization performance across four datasets compared to FedIT, as shown in Table 2. Interestingly, we observe that FedOA outperforms FedIT in OOD generalization for the majority of tasks, likely due to the fact that learning one task would enhance the performance of other tasks with shared underlying knowledge, whereas tasks that vary enormously may lead to degraded performance when learned together (Wei et al., 2021).

## 5.3 ANALYSIS

**Convergence analysis.** To analyze the convergence of different methods, we examine their average test accuracy versus communication rounds and present the OOD performance comparison on Reading Comprehension in Figure 1. As shown in Figure 1, our method exhibits a convergence speed comparable to other personalized methods, achieving notable performance enhancements after five communication rounds. This aligns with the discussion in Section 4, where FedOA could possess good convergence speed when appropriate learning step sizes are employed. The similar trends observed between our method and FedIT can be attributed to the benefit of feature distance-based regularization from the global adapter for OOD generalization.

Table 3: Ablation study of hyperparameter $\lambda$. RC represents the reading comprehension task.

| $\lambda$ | 0.01 | 0.1 | 0.5 | 1 | 2 |
|---|---|---|---|---|---|
| RC | 61.14 | 66.16 | 67.61 | 69.05 | 69.90 |

Table 4: Ablation study of different distance function $D$. RC represents reading comprehension task.

| $D$ | Cosine | Pearson | L2 |
|---|---|---|---|
| RC | 51.16 | 54.02 | 67.61 |

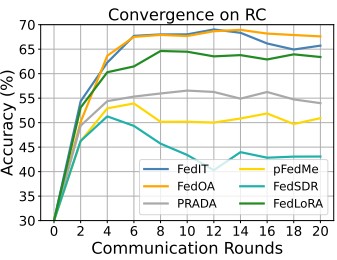

Figure 1: Average accuracy varies as communication rounds on reading comprehension task.

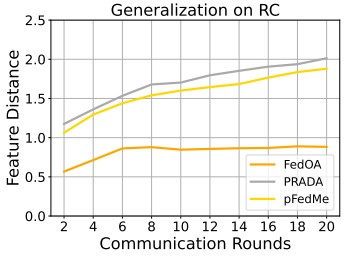

Figure 2: Feature distance between personalized and global models vs communication rounds.

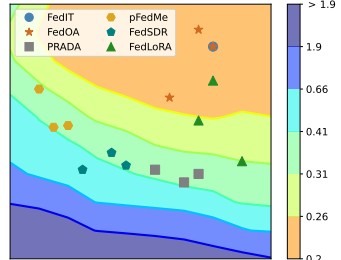

Figure 3: Loss surfaces w.r.t. model parameters on reading comprehension task.

**Generalization analysis.** Figure 3 visualizes the loss surfaces on the test environment for Reading Comprehension, using FedIT's global model as an anchor to position other personalized models. Compared with other methods, FedOA achieves better OOD generalization, as personalized models converge in flatter regions of the loss surface, supporting our theoretical motivation that reducing the distance between global and personalized model features leads to tighter generalization bounds. Additionally, the smaller gaps between global and personalized models highlight FedOA's advantage in maintaining a consistent optimization objective across clients, which is crucial for handling heterogeneous data across diverse domains. Figure 2 compares different regularization terms (the feature distance-based regularization of FedOA and the parameter regularization of pFedMe and PRADA) based on the average feature distances between personalized models and the global model. FedOA consistently maintains smaller and more stable feature distances, whereas the distances in other methods progressively increase. This aligns with the analysis in Section 3.2 and results in Table 2, demonstrating the effectiveness of our feature distance-based regularization approach.

**Sensitivity of $\lambda$.** In this study, we investigated the influence of the hyperparameter $\lambda$ during FedOA training with its value $\lambda \in \{0.01, 0.1, 0.5, 1, 2\}$. As shown in Table 3, increasing the regularization weight $\lambda$ will improve the OOD generalization performance, which can be attributed to the greater emphasis on aligning invariant features between the personalized and global models as the regularization strength increases. Notably, even with $\lambda = 0.1$, our proposed FedOA achieves superior performance compared to other baselines, which demonstrates the efficiency of our method.

**Effects of different distance function $D$.** To explore the impact of $D$, we conducted experiments of FedOA with Cosine, Pearson and L2- distance. As shown in Table 4, the L2-distance outperforms the others, demonstrating its effectiveness in feature distance calculation. Therefore, we choose the L2-distance function for our feature distance-based regularization during the training of FedOA.

## 6 CONCLUSION

FedFM offers a promising approach to enhancing foundation models using private data sources, but OOD generalization remains a critical challenge for the FedFM's application across diverse downstream tasks. Previous OOD methods in conventional FL are suboptimal for FedFM due to large parameter scale and increased data heterogeneity. To address these challenges, we begin with a theoretical generalization analysis of FedFM and propose an adapter-based method that incorporates feature distance-based regularization to improve OOD generalization in FedFM, simultaneously providing theoretical convergence guarantees. Our method is evaluated on public NLP tasks simulating an OOD FedFM setting. This work lays the foundation for addressing OOD generalization in FedFM, with future efforts focusing on more advanced methods and larger-scale settings.

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

## A  APPENDIX

The Appendix is organized as follows:

- Appendix  B provides related works.
- Appendix  C provides detailed dataset and baseline setups for experiments.
- Appendix  D provides generalization analysis of FedOA and the full proofs for Theorem 2.
- Appendix  E provides the convergence analysis of FedOA and the full proofs for Theorem 3.
- Appendix  F provides additional experiments demonstrating scalability and adaptability.

## B  RELATED WORK

### B.1  OUT-OF-DISTRIBUTION GENERALIZATION

Out-of-distribution (OOD) generalization addresses scenarios where the distribution of test data differs from that of the training data, a challenge that is critical for the successful deployment of models in real-world applications (Liu et al., 2021b; Arjovsky, 2020). Extensive research has focused on OOD generalization, exploring various assumptions and methodologies. For example, robust optimization methods (Namkoong & Duchi, 2016; Sagawa et al., 2019; Konstantinov & Lampert, 2019) aim to directly tackle the OOD generalization problem by optimizing for the worst-case error over a set of uncertainty distributions, with constrained relationships between training and testing environments. Causal learning methods (Gamella & Heinze-Deml, 2020; Oberst et al., 2021; Yang et al., 2021) draw upon concepts from causal inference to identify and leverage the underlying causal structure of the data, enabling prediction of the outcome variable based on these causal factors. Similarly, invariant learning (Arjovsky et al., 2019; Koyama & Yamaguchi, 2020; Liu et al., 2021a) seeks to identify and utilize the underlying heterogeneity and invariant representations or models across different environments by leveraging contextual information.

### B.2  GENERALIZATION IN FL

Recently, FL has emerged as a promising approach for utilizing private data in model training, prompting increased research into OOD generalization within the FL context (Li et al., 2023a; Yuan et al., 2021). Within this framework, a prevalent approach for achieving OOD generalization in FL is the adaptation of invariant learning based on representation learning. For instance, some studies (Zhang et al., 2021; Nguyen et al., 2022; Tan et al., 2024) employ feature alignment via adversarial/contrastive learning or regularization to align distributions across different clients, facilitating the learning of invariant representations. Similarly, other researchers (Guo et al., 2023; Tang et al., 2023) have adapted invariant risk minimization to develop representations that remain invariant to environment-specific variations while retaining relevance for the task at hand. Additionally, given the importance of robust aggregation in FL, numerous studies (Deng et al., 2020; Zhang et al., 2023b) have focused on improving aggregation algorithms to enhance OOD generalization.

Due to the increasing demand for personalized solutions in FL, recent research has focused on personalized federated learning (PFL) (Tan et al., 2022), which aims to learn an additional personalized model (T Dinh et al., 2020; Li et al., 2021a;b) or apply additional personalization steps (Fallah et al., 2020; Collins et al., 2021) to better align with individual user preferences. However, recent studies (Jiang & Lin, 2023; Ramasesh et al., 2021) have revealed that the personalized models in PFL can be prone to catastrophic forgetting and overfitting to local data, thus sacrificing their generalizability. Recent efforts have addressed these challenges by employing techniques such as regularization (Zhou et al., 2023; Xie et al., 2024) and designed structure for optimal classifiers (Chen & Chao, 2021; Luo et al., 2022; Li et al., 2023b), but these primarily focus on in-distribution generalization, where only seen training environments are considered during testing. This leaves OOD generalization as a significant unresolved issue in Personalized FL, particularly in the context of FedFM, where models are required to handle various downstream tasks in highly diverse and unseen environments. To fill this gap, we investigate the OOD generalization problem within the context of Federated Foundation Models, which are challenged by the substantial computational demands of large parameters and increased data heterogeneity.

### B.3 Federated Foundation Models

With the advent of foundation models, there has been a growing interest in integrating these models within the FL setting (Zhuang et al., 2023; Yu et al., 2023; Ren et al., 2024; Charles et al., 2024). In particular, due to the inherent computational and communication costs, recent research (Kuang et al., 2023; Zhang et al., 2023c) has focused on incorporating adapter-based parameter-efficient tuning (PEFT) methods with federated foundation models. Building on these efforts, numerous studies have emerged to address the challenges of integrating federated foundation models with adapter-based PEFT methods.

One notable contribution (Zhang et al., 2023a) pioneered the integration of instruction tuning within federated LLM frameworks. To tackle heterogeneity issues, previous works (Babakniya et al., 2023; Cho et al., 2023; Sun et al., 2024) introduced novel aggregation and initialization methods for LoRA to enhance the suitability of these models in FL environments. To further optimize the communication and computational overheads of FedFM, other research (Xu et al., 2023; Sun et al., 2023; Xu et al., 2024) has advanced gradient-free optimization techniques that are particularly well-suited for devices with limited memory and computational power. For personalization, one study (Yi et al., 2023) designed a specialized training paradigm for LoRA (Hu et al., 2021) to achieve more effective personalization in visually heterogeneous model scenarios. Additionally, another work (Yang et al., 2024) proposed a dual-adapter framework that incorporates an additional personalized model to enhance personalization efforts. Regarding generalization, a pioneering study (Du et al., 2024) was the first to investigate the generalization degradation that occurs when directly tuning foundation models in FL via robustness analysis experiments. Diverging from these approaches, our work explores the OOD generalization problem in FedFM through comprehensive theoretical analysis, extending the scope of research in this area.

## C Implementation Details

### C.1 Datasets

In this paper, we developed four datasets derived from the Flan (Wei et al., 2021), and details of their construction are elucidated in this section. Flan comprises a diverse range of NLP tasks, each containing multiple datasets from different domains. To align with OOD settings, we employed a stratified selection process, choosing four distinct tasks to represent four environments and randomly selecting two datasets with different sources for each task from Flan. To simulate client local data scarcity (McMahan et al., 2017), we applied a downsampling strategy, reducing each selected local dataset to 1000 training instances and 200 testing instances. In experiments, we employed a "leave-one-task-out" strategy, setting aside one task as the test environment while using the remaining tasks as training environments. For example, if the task of Entailment (comprising test instances from the snli and anli datasets) is selected as the test dataset, then the remaining six datasets of three tasks (Sentiment, Paraphrase and Reading Comprehension) are used for training with each client contains one dataset. Consequently, each tested federated OOD dataset encompasses three distinct NLP tasks, with two datasets for each task, yielding a total of 6000 training examples and 1200 testing examples. The specific tasks and datasets included are listed in Table 5.

### C.2 Baselines and Implementation

In this section, detailed descriptions of the implementation of each baseline compared in this study will be provided:

- **Centralized model:** This model is trained by gathering data from all training environments into a single centralized framework, with 10 epochs to optimize.

- **FedIT (Zhang et al., 2023a):** FedIT extends FedAVG (McMahan et al., 2017) to foundation models by incorporating LoRA tuning for instruction learning. After training on diverse local client datasets, the final aggregated global model is utilized for testing.

- **pFedMe (T Dinh et al., 2020):** pFedMe learns personalized models through Moreau envelopes regularization. To ensure a fair comparison, we adapt pFedMe to the FedFM set-

Table 5: Tasks and datasets included in the constructed federated OOD datasets.

| Tasks | Datasets | Sources |
|---|---|---|
| Entailment | snli | Captions |
| | anli | Wikipedia, WikiHow, news, fiction and formal spoken text |
| Sentiment | sst2 | Movie reviews |
| | sentiment140 | Tweets |
| Paraphrase | glue_mrpc | Newswire articles |
| | stsb | News headlines, captions and NLI data |
| Reading Comprehension | openbook qa | Wikipedia and ConceptNet |
| | record | CNN/Daily Mail news articles |

ting by incorporating adapter tuning, where only the adapter parameters are learned and regularization is applied specifically to the adapters.

- **FedLoRA (Yi et al., 2023):** FedLoRA incorporates LoRA for efficient learning in model-heterogeneous settings and employs additional local tuning as a personalized adaptation process. Here, we adapt the training paradigm in FedLoRA to NLP tasks, utilizing the personalized LoRAs for testing. These personalized LoRAs are derived through further local tuning on each client's dataset after obtaining the globally aggregated LoRA.

- **PERADA (Xie et al., 2024):** PERADA utilizes adapters for efficient learning and applies adapter parameter regularization to improve the generalization capability of the personalized model. In this work, we adapt PERADA to the FedFM framework for NLP tasks, excluding the distillation of the global adapter.

- **FedSDR (Tang et al., 2023):** FedSDR aims to learn optimal personalized causally invariant predictors through conditional mutual information regularization for addressing OOD scenrios in FL. In this work, we adapt pFedMe to the FedFM setting by incorporating adapter tuning, where only the adapter parameters are learned and regularization is applied specifically to the adapters. Additionally, due to the fixed head in foundation model tuning, we omit the head regularization component typically used for shortcut extractor learning in FedSDR.

All models are implemented using LoRA to enhance learning efficiency, with the rank of LoRA set as $r = 8$ and only applied to $W_q$ and $W_v$. For FL methods, each client conducts $K = 2$ local epochs with a batch size of 32. We implement all the methods using PyTorch and conduct all experiments on NVIDIA A40 GPUs.

## D GENERALIZATION ANALYSIS

We first analyze the generalization bound of the conventional aggregated global model. We define the aggregated global hypothesis $f_g$ with its objective as $f_g = \arg\min_{f \in \mathcal{F}} \sum_{e \in \mathcal{E}_{train}} \alpha_e R_e(f)$. Following previous work (Konstantinov & Lampert, 2019), we can get the upper bound of risk of the global hypothesis $f_g$ as Lemma 1.

**Lemma 1.** (Generalization bound of aggregated global). *Let $f_e^* = \arg\min_{f \in \mathcal{F}} \mathcal{R}_e(f)$ and assume that $\ell(.,.) \leq M$, then for any $e \in \mathcal{E}_{all}$ and $\delta > 0$, with probability at least $1 - \delta$ over the data, the excess risk of the learned global model $f_g$ can be bounded by:*

$$\mathcal{R}_e(f_g) \leq \mathcal{R}_e(f_e^*) + \sum_{e' \in \mathcal{E}_{train}} \alpha_{e'} H_{e'}(\mathcal{F}) + 2 \sum_{e' \in \mathcal{E}_{train}} \alpha_{e'} d_{\mathcal{F}}(P_e, P_{e'}) + C \sqrt{\sum_{e' \in \mathcal{E}_{train}} \frac{\alpha_{e'}}{|S_{e'}|}} \quad (9)$$

*where, $C = 6\sqrt{\frac{\log(\frac{4}{\delta})M^2}{2}}$, for each client $e$, $H_e(\mathcal{F})$ is the empirical Rademacher complexity $\mathcal{F}$ and $d_{\mathcal{F}}(P_e, P_{e'})$ is the discrepancy between the distributions $P_e$ and $P_{e'}$ with hypothesis class $\mathcal{F}$, defined as:*

$$d_{\mathcal{F}}(P_e, P_{e'}) = Supp_{f \in \mathcal{F}}(|\mathcal{R}_e(f) - \mathcal{R}_{e'}(f)|) \quad (10)$$

Following previous work (Guo et al., 2023), we have the definition of invariant predictor (a model only uses invariant features to predict) as Definition 1.

**Definition 1.** (Invariant Predictor). If there is a head $w$ simultaneously optimal for all environments $w \in \arg\min_w R_e(w, \Phi)$ for all $e \in \mathcal{E}_{all}$, the invariant predictor $f = (w, \Phi)$ is elicited based on the representation $\Phi$.

**Proof of Theorem 1 (Conventional aggregated global model in FedFM inherently retains OOD generalization ability).** During tuning, the pre-trained head $w$ of foundation models is fixed and taken as the optimal head for all tasks (Hu et al., 2023). Therefore, the objective of global hypothesis $f_g$ can be further formalized as follows:

$$\min_{\Phi_g} \sum_{e \in \mathcal{E}_{train}} \alpha_e R_e(w, \Phi_g) \tag{11}$$
$$s.t. \quad w \in \arg\min_w R_e(w, \Phi_g), \text{ for all } e \in \mathcal{E}_{train}.$$

By omitting the pre-trained head, the objective of global hypothesis $f_g$ simplifies to $\min_{\Phi_g} \sum_{e \in \mathcal{E}_{train}} \alpha_e R_e(\Phi_g)$, aligning with objective (2) to learn invariant features that satisfy Assumption 1, according to Definition 1. Hence, based on Lemma 1, when Assumption 1 holds, the discrepancy in the generalization bound of the global hypothesis $f_g$ in federated foundation models approaches zero $d_{\mathcal{F}}(P_e, P_{e'}) = Supp_{f \in \mathcal{F}}(|\mathcal{R}_e(f) - \mathcal{R}_{e'}(f)|) = Supp_{f \in \mathcal{F}}(|\mathbb{E}[\ell(w(z)), Y^e] - \mathbb{E}[\ell(w(z)), Y^{e'}]|) \to 0$, and is more tightly bounded by the representation $\Phi$ during learning $d_{\mathcal{F}}(P_e, P_{e'}) = Supp_{f \in \mathcal{F}}(|\mathcal{R}_e(f) - \mathcal{R}_{e'}(f)|) = Supp_{\cup \Phi}(|\mathcal{R}_e(\Phi) - \mathcal{R}_{e'}(\Phi)|)$.

Next, we provide proof of Theorem 2, where local hypothesis is $f_e = (w, \Phi_e)$ and global hypothesis is $f_g = (w, \Phi_g)$.

**Theorem 2.** (Generalization bound of personalized model). *Assume that $\ell(.,.) \leq M$ and $f_e^* = \arg\min_{f \in \mathcal{F}} \mathcal{R}_e(f)$, then for any $e \in \mathcal{E}_{all}$ and $\delta > 0$, with probability at least $1 - \delta$ over the data, the excess risk of the learned personalized model $f_e$ can be bounded by:*

$$\mathcal{R}_e(f_e) \leq \mathcal{R}_e(f_e^*) + M \cdot \mathbb{E}_{X \sim P_e}[D(\Phi_e(X), \Phi_g(X))] + \sum_{e' \in \mathcal{E}_{train}} \alpha_{e'} H_{e'}(\mathcal{F})$$
$$+ 2 \sum_{e' \in \mathcal{E}_{train}} \alpha_{e'} d_{\mathcal{F}}(P_e, P_{e'}) + C \sqrt{\sum_{e' \in \mathcal{E}_{train}} \frac{\alpha_{e'}}{|S_{e'}|}} \tag{12}$$

Proof.

$$\mathcal{R}_e(f_e) = \underbrace{\mathcal{R}_e(f_e) - \mathcal{R}_e(f_g)}_{A_1} + \mathcal{R}_e(f_g) \tag{13}$$

Assume $z = \Phi(x)$, for the first term $A_1$, we have

$$A_1 = \mathbb{E}_{z \sim P(\Phi_e(X))}[\mathbb{E}_{y \sim P(Y|Z=z)}[\ell(w(z), y)]] - \mathbb{E}_{z' \sim P(\Phi_g(X))}[\mathbb{E}_{y \sim P(Y|Z=z')}[\ell(w(z), y)]]$$
$$= \mathbb{E}_{z \sim P(\Phi_e(X))}[\mathbb{E}_{y \sim P(Y|Z=z)}[\ell(w(z), y)]] - \mathbb{E}_{z \sim P(\Phi_e(X))}[\mathbb{E}_{y \sim P(Y|Z=z')}[\ell(w(z), y)]]$$
$$+ \mathbb{E}_{z \sim P(\Phi_e(X))}[\underbrace{\mathbb{E}_{y \sim P(Y|Z=z')}[\ell(w(z), y)]}_{g(z)}] - \mathbb{E}_{z' \sim P(\Phi_g(X))}[\underbrace{\mathbb{E}_{y \sim P(Y|Z=z')}[\ell(w(z), y)]}_{g(z)}]$$
$$\overset{(a)}{\leq} \mathbb{E}_{z \sim P(\Phi_e(X))}[g(z)] - \mathbb{E}_{z' \sim P(\Phi_g(X))}[g(z)]$$
$$\overset{(b)}{\leq} M \cdot \mathbb{E}_{X \sim P_e}[D(\Phi_e(X), \Phi_g(X))] \tag{14}$$

where (a) is from Assumption 1, (b) is from the condition that $|g(z)| \leq M$ if $\ell(.,.) \leq M$, and $D$ represents a function to measure distance.

Plugging back the bounds on $A_1$ and Lemma 1, obtaining

$$\mathcal{R}_e(f_e) \leq \mathcal{R}_e(f_e^*) + M \cdot \mathbb{E}_{X \sim P_e}[D(\Phi_e(X), \Phi_g(X))] + \sum_{e' \in \mathcal{E}_{train}} \alpha_{e'} H_{e'}(\mathcal{F})$$
$$+ 2 \sum_{e' \in \mathcal{E}_{train}} \alpha_{e'} d_{\mathcal{F}}(P_e, P_{e'}) + C \sqrt{\sum_{e' \in \mathcal{E}_{train}} \frac{\alpha_{e'}}{|S_{e'}|}} \tag{15}$$

# E    CONVERGENCE ANALYSIS

## E.1    TECHNICAL LEMMAS

We first present some technical lemmas involved in later proofs, where Lemma 2 and Lemma 3 can be found in (Karimireddy et al., 2020) and (T Dinh et al., 2020), respectively.

**Lemma 2.** (Relaxed triangle inequality). *For any vectors $v_1, v_2 \in \mathbb{R}^d$ and $a > 0$, we have*

$$||v_1 + v_2||^2 \leq (1 + a)||v_1||^2 + (1 + \frac{1}{a})||v_2||^2. \tag{16}$$

**Lemma 3.** (Relaxed triangle inequality). *For any $x \in \mathbb{R}, n \in \mathbb{N}$, we have*

$$\sum_{i=0}^{N-1} x^i = \frac{x^n - 1}{x - 1},$$
$$(1 + \frac{x}{n})^n \leq e^x \tag{17}$$

**Lemma 4.** (Heterogenity Bound). *Suppose that Assumption 4 holds true, we have*

$$\mathbb{E}||\nabla R(\phi)||^2 \leq 2\mathbb{E}||\nabla R_e(\phi)||^2 + 2G^2 \tag{18}$$

Proof. Using Lemma 2 and Assumption 4, we have

$$\begin{aligned} \mathbb{E}||\nabla R(\phi)||^2 &= \mathbb{E}||\nabla R(\phi) - \nabla R_e(\phi) + \nabla R_e(\phi)||^2 \\ &\leq 2\mathbb{E}||\nabla R_e(\phi)||^2 + 2G^2 \end{aligned} \tag{19}$$

## E.2    CONVERGENCE RESULTS

In this section, we provide proof of Theorem 3, focusing exclusively on the small tunable parameter $\phi$, while disregarding the frozen parameters.

We begin by defining the local updates for each client $e$. The client's global model, with parameter $\phi_g^{t-1}$, and the personalized model, initialized with $\phi_{e,0}^t = \phi_e^{t-1}$, undergo K local updates with L2-distance function $D$, as follows:

$$\begin{aligned} \phi_{e,k}^t &= \phi_{e,k-1}^t - \eta_l g_e(\phi_{e,k-1}^t, \phi_g^{t-1}) \\ &= \phi_{e,k-1}^t - \eta_l[\nabla R_e(\phi_{e,k-1}^t; \xi) + \lambda \nabla D(\Phi(\phi_{e,k-1}^t; \xi), \Phi(\phi_g^{t-1}; \xi))] \\ &= \phi_{e,k-1}^t - \eta_l[\nabla R_e(\phi_{e,k-1}^t; \xi) + 2\lambda \nabla \Phi(\phi_{e,k-1}^t; \xi)|\Phi(\phi_{e,k-1}^t; \xi) - \Phi(\phi_g^{t-1}; \xi)]] \end{aligned} \tag{20}$$

We then bound the client drift error.

**Lemma 5.** *Suppose that Assumption 2 and 3 hold true, our method updates with constant local step-size such that $\eta_l \leq \frac{1}{4\sqrt{2(1+2K)K\lambda\sigma L}}$. Then, for all $t \in [T]$, we can bound the client drift error as follows:*

$$\mathbb{E}||\phi_{e,K}^t - \phi_{e,0}^t||^2 \leq 32K(1 + 2K)\lambda^2\sigma^2 L^2\eta_l^2 \mathbb{E}||\phi_{e,0}^t - \phi_g^{t-1}||^2 + 4K(1 + 2K)\sigma^2\eta_l^2 \tag{21}$$

Proof.

$$\begin{aligned} \mathbb{E}||\phi_{e,K}^t - \phi_{e,0}^t||^2 &= \mathbb{E}||\phi_{e,K-1}^t - \phi_{e,0}^t - \eta_l g_c(\phi_{e,K-1}^t, \phi_g^{t-1})||^2 \\ &\overset{(a)}{\leq} (1 + \frac{1}{2K})\mathbb{E}||\phi_{e,K-1}^t - \phi_{e,0}^t||^2 + \underbrace{(1 + 2K)\eta_l^2\mathbb{E}||g_c(\phi_{e,K-1}^t, \phi_g^{t-1})||^2}_{A_1} \end{aligned} \tag{22}$$

where (a) is from Lemma 2 with $a = 2K$. For the second term, we have

$$
\begin{aligned}
A_1 =& (1+2K)\eta_l^2 \mathbb{E}||\nabla R_e(\phi_{e,K-1}^t;\xi) + 2\lambda\nabla\Phi(\phi_{e,K-1}^t;\xi)||\Phi(\phi_{e,K-1}^t;\xi) - \Phi(\phi_g^{t-1};\xi)||||^2 \\
&\overset{(b)}{\leq} 2(1+2K)\eta_l^2 \mathbb{E}||\nabla R_e(\phi_{e,K-1}^t;\xi)||^2 \\
&\quad + 8(1+2K)\lambda^2\eta_l^2 \mathbb{E}[||\nabla\Phi(\phi_{e,K-1}^t;\xi)||^2 \cdot ||\Phi(\phi_{e,K-1}^t;\xi) - \Phi(\phi_g^{t-1};\xi)||^2] \\
&\overset{(c)}{\leq} 2(1+2K)\sigma^2\eta_l^2 + 8(1+2K)\lambda^2\sigma^2 L^2\eta_l^2 \mathbb{E}||\phi_{e,K-1}^t - \phi_g^{t-1}||^2 \\
&\overset{(d)}{\leq} 2(1+2K)\sigma^2\eta_l^2 + 16(1+2K)\lambda^2\sigma^2 L^2\eta_l^2 \mathbb{E}||\phi_{e,K-1}^t - \phi_{e,0}^t||^2 \\
&\quad + 16(1+2K)\lambda^2\sigma^2 L^2\eta_l^2 \mathbb{E}||\phi_{e,0}^t - \phi_g^{t-1}||^2
\end{aligned}
\tag{23}
$$

where (b) is from Lemma 2 with $a = 1$, (c) is from Assumption 2 and Assumption 3, and (d) is from Lemma 2 with $a = 1$. Plugging back the bounds on $A_1$, we obtain the recursive bound of the client drift error:

$$
\begin{aligned}
\mathbb{E}||\phi_{e,K}^t - \phi_{e,0}^t||^2 \leq& (1 + \frac{1}{2K} + 16(1+2K)\lambda^2\sigma^2 L^2\eta_l^2)\mathbb{E}||\phi_{e,K-1}^t - \phi_{e,0}^t||^2 \\
&+ 16(1+2K)\lambda^2\sigma^2 L^2\eta_l^2 \mathbb{E}||\phi_{e,0}^t - \phi_g^{t-1}||^2 + 2(1+2K)\sigma^2\eta_l^2 \\
&\overset{(e)}{\leq} (1 + \frac{1}{K})\mathbb{E}||\phi_{e,K-1}^t - \phi_{e,0}^t||^2 + 16(1+2K)\lambda^2\sigma^2 L^2\eta_l^2 \mathbb{E}||\phi_{e,0}^t - \phi_g^{t-1}||^2 \\
&\quad + 2(1+2K)\sigma^2\eta_l^2 \\
&\overset{(f)}{\leq} (16(1+2K)\lambda^2\sigma^2 L^2\eta_l^2 \mathbb{E}||\phi_{e,0}^t - \phi_g^{t-1}||^2 + 2(1+2K)\sigma^2\eta_l^2) \sum_{i=0}^{K-1}(1 + \frac{1}{K})^i \\
&\overset{(g)}{\leq} 32K(1+2K)\lambda^2\sigma^2 L^2\eta_l^2 \mathbb{E}||\phi_{e,0}^t - \phi_g^{t-1}||^2 + 4K(1+2K)\sigma^2\eta_l^2
\end{aligned}
\tag{24}
$$

where (e) is from the condition on local step-size that $\eta_l \leq \frac{1}{4\sqrt{2(1+2K)K}\lambda\sigma L}$ implying that $16(1 + 2K)\lambda^2\sigma^2 L^2\eta_l^2 \leq \frac{1}{2K}$, (f) is from the unrolling recursion, and (g) is from Lemma 3 with $\sum_{i=0}^{K-1}(1+\frac{1}{K})^i = \frac{(1+1/K)^K - 1}{1/K} \leq \frac{e-1}{1/K} \leq 2K$.

**Lemma 6.** *Suppose that Assumption 2, 3 and 4 hold true, our method updates with constant local and global step-size such that $\eta_l \leq \frac{1}{8\sqrt{3(1+3T)T(1+2K)K}\lambda\sigma L}$ and $\eta_g \leq \frac{1}{2\sqrt{6(1+3T)T}L}$. Then, we have:*

$$
\mathbb{E}||\phi_e^t - \phi_g^t||^2 \leq 3\mathbb{E}||\phi_e^0 - \phi_g^0||^2 + 16(1+3T)TK(1+2K)\sigma^2\eta_l^2 + 8(1+3T)T\eta_g^2 G^2
\tag{25}
$$

Proof.

$$
\begin{aligned}
\mathbb{E}||\phi_e^t - \phi_g^t||^2 =& \mathbb{E}||\phi_e^{t-1} - \phi_g^{t-1} + \phi_e^t - \phi_e^{t-1} + \phi_g^{t-1} - \phi_g^t||^2 \\
&\overset{(a)}{\leq} (1 + \frac{1}{3T})\mathbb{E}||\phi_e^{t-1} - \phi_g^{t-1}||^2 + \underbrace{(1+3T)\mathbb{E}||\phi_e^t - \phi_e^{t-1} + \phi_g^{t-1} - \phi_g^t||^2}_{A_1}
\end{aligned}
\tag{26}
$$

where (a) is from Lemma 2 with $a = 3T$. For the second term, we have

$$
\begin{aligned}
A_1 =& (1+3T)\mathbb{E}||\phi_e^t - \phi_e^{t-1} + \phi_g^{t-1} - \phi_g^t||^2 \\
&\overset{(b)}{\leq} 2(1+3T)\mathbb{E}||\phi_e^t - \phi_e^{t-1}||^2 + 2(1+3T)\eta_g^2 \mathbb{E}||\nabla R(\phi_g^{t-1})||^2 \\
&\overset{(c)}{\leq} 2(1+3T)\mathbb{E}||\phi_e^t - \phi_e^{t-1}||^2 + 4(1+3T)\eta_g^2 \mathbb{E}||\nabla R_e(\phi_g^{t-1})||^2 + 4(1+3T)\eta_g^2 G^2 \\
&\overset{(d)}{\leq} 2(1+3T)\mathbb{E}||\phi_e^t - \phi_e^{t-1}||^2 + 8(1+3T)\eta_g^2 \mathbb{E}||\nabla R_e(\phi_g^{t-1}) - \nabla R_e(\phi_e^{t-1})||^2 \\
&\quad + 8(1+3T)\eta_g^2 \mathbb{E}||\nabla R_e(\phi_e^{t-1})||^2 + 4(1+3T)\eta_g^2 G^2 \\
&\overset{(e)}{\leq} 64(1+3T)K(1+2K)\lambda^2\sigma^2 L^2\eta_l^2 \mathbb{E}||\phi_e^{t-1} - \phi_g^{t-1}||^2 + 8(1+3T)K(1+2K)\sigma^2\eta_l^2 \\
&\quad + 8(1+3T)L^2\eta_g^2 \mathbb{E}||\phi_e^{t-1} - \phi_g^{t-1}||^2 + 8(1+3T)\sigma^2\eta_g^2 + 4(1+3T)\eta_g^2 G^2
\end{aligned}
\tag{27}
$$

where (b) is from Lemma 2 with $a = 1$, (c) is from Lemma 4, (d) is from Lemma 2 with $a = 1$, (e) is from Lemma 5 with $\phi_e^{t-1} = \phi_{e,0}^t, \phi_e^t = \phi_{e,K}^t$ and Assumption 2 and Assumption 3. Plugging back the bounds on $A_1$, we obtain the recursive bound as:

$$
\begin{aligned}
\mathbb{E}||\phi_e^t - \phi_g^t||^2 \leq & (1 + \frac{1}{3T})\mathbb{E}||\phi_e^{t-1} - \phi_g^{t-1}||^2 + 64(1+3T)K(1+2K)\lambda^2\sigma^2 L^2\eta_l^2\mathbb{E}||\phi_e^{t-1} - \phi_g^{t-1}||^2 \\
& + 8(1+3T)K(1+2K)\sigma^2\eta_l^2 + 8(1+3T)L^2\eta_g^2\mathbb{E}||\phi_e^{t-1} - \phi_g^{t-1}||^2 \\
& + 8(1+3T)\sigma^2\eta_g^2 + 4(1+3T)\eta_g^2 G^2 \\
\overset{(f)}{\leq} & (1 + \frac{1}{T})\mathbb{E}||\phi_e^{t-1} - \phi_g^{t-1}||^2 + 8(1+3T)K(1+2K)\sigma^2\eta_l^2 + 4(1+3T)\eta_g^2 G^2 \\
\overset{(g)}{\leq} & (8(1+3T)K(1+2K)\sigma^2\eta_l^2 + 4(1+3T)\eta_g^2 G^2)\sum_{i=0}^{T-1}(1 + \frac{1}{T})^i + (1 + \frac{1}{T})^T\mathbb{E}||\phi_e^0 - \phi_g^0||^2 \\
\overset{(h)}{\leq} & 3\mathbb{E}||\phi_e^0 - \phi_g^0||^2 + 16(1+3T)TK(1+2K)\sigma^2\eta_l^2 + 8(1+3T)T\eta_g^2 G^2
\end{aligned}
$$
(28)

where (f) is from the condition on global step-size that $\eta_g \leq \frac{1}{2\sqrt{6(1+3T)TL}}$ implying that $8(1 + 3T)L^2\eta_g^2 \leq \frac{1}{3T}$, and local step-size that $\eta_l \leq \frac{1}{8\sqrt{3(1+3T)T(1+2K)K}\lambda\sigma L}$ implying that $64(1+3T)K(1+2K)\lambda^2\sigma^2 L^2\eta_l^2 \leq \frac{1}{3T}$, (g) is from the unrolling recursion, and (h) is from Lemma 3.

Next, we prove the progress made in each round.

**Lemma 7.** *Suppose that Assumption 2, 3 and 4 hold true, our method updates with constant local and global step-size such that $\eta_l \leq \frac{1}{8\sqrt{3(1+3T)T(1+2K)K}\lambda\sigma L}$ and $\eta_g \leq \frac{1}{2\sqrt{6(1+3T)TL}}$. Then, our method makes progress in each round as follows:*

$$
\begin{aligned}
\mathbb{E}R_e(\phi_e^t) \leq & \mathbb{E}R_e(\phi_e^{t-1}) - \frac{1}{2}||\nabla R_e(\phi_e^{t-1})||^2 + 48K(1+2K)\lambda^2\sigma^2(L-1)L^2\eta_l^2 M^2 \\
& + 128K(1+2K)T(1+3T)\lambda^2\sigma^2(L-1)L^2 G^2\eta_l^2\eta_g^2 + 4K(1+2K)(L-1)\sigma^2\eta_l^2
\end{aligned}
$$
(29)

Proof. Starting from the smoothness, we have

$$
\begin{aligned}
\mathbb{E}R_e(\phi_e^t) \leq & \mathbb{E}R_e(\phi_e^{t-1}) + \mathbb{E}\langle\nabla R_e(\phi_e^{t-1}), \phi_e^t - \phi_e^{t-1}\rangle + \frac{L}{2}\mathbb{E}||\phi_e^t - \phi_e^{t-1}||^2 \\
\overset{(a)}{\leq} & \mathbb{E}R_e(\phi_e^{t-1}) + \frac{L}{2}\mathbb{E}||\phi_e^t - \phi_e^{t-1}||^2 - \frac{1}{2}||\nabla R_e(\phi_e^{t-1})||^2 - \frac{1}{2}\mathbb{E}||\phi_e^t - \phi_e^{t-1}||^2 \\
\overset{(b)}{\leq} & \mathbb{E}R_e(\phi_e^{t-1}) - \frac{1}{2}||\nabla R_e(\phi_e^{t-1})||^2 + 16K(1+2K)\lambda^2\sigma^2(L-1)L^2\eta_l^2\mathbb{E}||\phi_e^{t-1} - \phi_g^{t-1}||^2 \\
& + 2K(1+2K)(L-1)\sigma^2\eta_l^2 \\
\overset{(c)}{\leq} & \mathbb{E}R_e(\phi_e^{t-1}) - \frac{1}{2}||\nabla R_e(\phi_e^{t-1})||^2 + 48K(1+2K)\lambda^2\sigma^2(L-1)L^2\eta_l^2 M^2 \\
& + 128K(1+2K)T(1+3T)\lambda^2\sigma^2(L-1)L^2 G^2\eta_l^2\eta_g^2 + 4K(1+2K)(L-1)\sigma^2\eta_l^2
\end{aligned}
$$
(30)

where (a) is from that $-\langle\boldsymbol{a}, \boldsymbol{b}\rangle \leq \frac{1}{2}(||\boldsymbol{a}||^2 + ||\boldsymbol{b}||^2)$, (b) is from that $\phi_{e,0}^t = \phi_e^{t-1}$ and substituting with Lemma 5, and (c) is from that $\mathbb{E}||\phi_e^0 - \phi^0||^2 \leq M^2$ and substituting with Lemma 6

Finally, we can get convergence results for the general non-convex case of our method.

**Theorem 3.** *Suppose that Assumption 2, 3 and 4 hold true, our method updates with constant local and global step-size such that $\eta_l \leq \frac{1}{8\sqrt{3(1+3T)T(1+2K)K}\lambda\sigma L}$ and $\eta_g \leq \frac{1}{2\sqrt{6(1+3T)TL}}$. Then, the sequence of iterates generated by our method satisfies:*

$$
\begin{aligned}
\frac{1}{T}\sum_{t=1}^{T}\mathbb{E}||\nabla R_e(\phi_e^{t-1})||^2 \leq & \frac{2(\mathbb{E}R_e(\phi_e^0) - \mathbb{E}R_e(\phi_e^*))}{T} + 8K(1+2K)(L-1)(1+12\lambda^2 L^2 M^2)\sigma^2\eta_l^2 \\
& + 256K(1+2K)T(1+3T)\lambda^2\sigma^2(L-1)L^2 G^2\eta_l^2\eta_g^2
\end{aligned}
$$
(31)

*If we choose the step sizes $\eta_l = \mathcal{O}(\frac{1}{TKL\sigma})$ and $\eta_g = \mathcal{O}(\frac{1}{TL})$, we have the convergence rates of our method as follows*

$$\frac{1}{T}\sum_{t=1}^{T}\mathbb{E}||\nabla R_e(\phi_e^{t-1})||^2 = \mathcal{O}\left(\frac{(\mathbb{E}R_e(\phi_e^0) - \mathbb{E}R_e(\phi_e^*))}{T}, \frac{1 + \lambda^2 L^2 M^2}{T^2 L}, \frac{\lambda^2 G^2}{T^2 L}\right) \qquad (32)$$

Proof. Summing up all the $T$ inequalities in Equation of Lemma 7, we have

$$\frac{1}{T}\sum_{t=1}^{T}\mathbb{E}||\nabla R_e(\phi_e^{t-1})||^2 \leq \frac{2\sum_{t=1}^{T}(\mathbb{E}R_e(\phi_e^{t-1}) - \mathbb{E}R_e(\phi_e^t))}{T} + 8K(1+2K)(L-1)(1+12\lambda^2 L^2 M^2)\sigma^2\eta_l^2$$

$$+ 256K(1+2K)T(1+3T)\lambda^2\sigma^2(L-1)L^2 G^2\eta_l^2\eta_g^2$$

$$\overset{(a)}{\leq} \frac{2(\mathbb{E}R_e(\phi_e^0) - \mathbb{E}R_e(\phi_e^*))}{T} + 8K(1+2K)(L-1)(1+12\lambda^2 L^2 M^2)\sigma^2\eta_l^2$$

$$+ 256K(1+2K)T(1+3T)\lambda^2\sigma^2(L-1)L^2 G^2\eta_l^2\eta_g^2$$

$$(33)$$

where (a) is from that $\mathbb{E}R_e(\phi_e^*) \leq \mathbb{E}R_e(\phi_e^T)$.

# F    ADDITIONAL EXPERIMENTS

## F.1    SCALABILITY ANALYSIS

To assess the scalability of our approach, we conducted experiments by increasing the number of clients to 30. We compared our method against the top two personalized methods and a global model method specifically on the Reading Comprehension task. The results, as detailed in Table 6, demonstrate that our method consistently outperforms the others, showcasing superior stability and effectiveness under expanded client scenarios. These findings highlight the potential of our approach to be effectively scaled, catering to more complex and larger federated settings while maintaining performance benchmarks.

Table 6: Ablation study of scalability with 30 clients. RC represents reading comprehension task. FedIT are tested on a single global model, while the remaining models are tested on personalized models with average results reported.

| $Methods$ | **FedIT** | **PRADA** | **FedLoRA** | **FedOA** |
|---|---|---|---|---|
| RC | 58.04 | 39.30 | 46.90 | **58.84** |

## F.2    ADAPTABILITY ANALYSIS

To enhance applicability across diverse non-IID environments, our method is strategically designed for high flexibility, enabling adaptation across various global learning frameworks, backbones and PEFT methods for different scenarios. This adaptability is simply achieved through the straightforward substitution of the FedAvg, LLM and LoRA with alternative aggregation methods, transformer-based foundation models and adapter-based PEFT methods during the training. In our experiment, we employ FedAvg, LLM and LoRA as representative examples, demonstrating our methods' superior performance compared to other baselines as indicated in Table 2.

To further validate the effectiveness and versatility of our approach across different federated foundation model contexts, we extend our methods to include the ViT (Dosovitskiy, 2020) framework and also implement other baselines within ViT to maintain a fair comparison. We conduct experiments on OfficeHome datatset (Venkateswara et al., 2017),which comprises images across four distinct domains with 65 categories. In line with our previous experiments, we employed a "leave-one-domain-out" strategy, where each of the three clients maintains data from one distinct domain, setting aside the remaining domain as the testing data for evaluating OOD generalization. Results

presented in Table 7 indicate that our methods outperform other personalized models and have comparable results with global models. These findings underscore the robustness and consistent efficacy of our methods across various federated foundation models context.

Table 7: OOD results of different models using "leave-one-domain-out" validation. FedIT are tested on a single global model, while the remaining models are tested on personalized models with average results reported.

| Methods | Art | CliPart | Product | Real World | Average |
|---------|-----|---------|---------|-----------|---------|
| FedIT | 68.11 | 56.66 | 77.18 | 77.94 | 69.97 |
| pFedMe | 54.72 | 41.25 | 59.22 | 60.67 | 53.96 |
| FedLoRA | 60.49 | 51.31 | 72.93 | 73.15 | 64.47 |
| PRADA | 54.73 | 41.25 | 59.24 | 60.68 | 53.98 |
| FedOA | **67.49** | **56.51** | **75.96** | **77.45** | **69.35** |

