# OpenReview forum: "Federated Adapter on Foundation Models:  An Out-Of-Distribution Approach"
_ICLR.cc/2025/Conference — Submitted to ICLR 2025_

### Official Review · Reviewer_DApR · 2024-10-24

**Soundness:** 2
**Presentation:** 3
**Contribution:** 1
**Rating:** 3
**Confidence:** 4

**Summary:**

This paper proposes FedOA to tackle the out-of-distribution issue in federated learning with foundation models (i.e., LLM in their narrative). Their method employs adapter-based parameter-efficient fine-tuning methods for efficient learning, and introduces an additional personalized model with a feature distance-based regularization to ensure distribution alignment and provide OOD generalization guarantees for each client. Some theoretical results are also presented.

**Strengths:**

- Federated learning with foundation models is meaningful, so the topic is interesting.
- The presentation of this paper is good, especially the figures.

**Weaknesses:**

- **Foundation models**. This paper mention that the method is for foundation models, however it is not well motivated and well justified.
  - The authors mentioned that _"A key distinction between FedFM and conventional FL lies in the scale of parameters involved"_, which I agree. But according to the theoretical analysis, I didn't see any assumptions or results that can be distinguished from conventional FL to reflect the "foundation model properties".
  - The authors mentioned that previous conventional methods cannot perform well in FedFM, however, the authors even didn't compare the conventional FL baselines in the experiments. As far as I concern, previous methods can be adapted in FedFM and also have good performances, a good reference is [1]. Instead of the full models, previous methods can be adopted in the adapter too.
  - The wording "foundation model" should be used in more careful and appropriate ways. If you are claiming your work is for foundation models, at least you should conduct experiments in various foundation models in both CV and NLP tasks with models like LLM, Diffusion models, Vision transformers, CLIP, etc. However, this paper only has experiments in LLaMA-2, which is not sufficient for LLM itself. Or, the authors can claim they target FedLLM, but they should clearly present the challenges and tailored methods for LLM instead of roughly speaking "foundation models."
- **Lack of important literature**. There are places in this paper where the authors' claims are not (or less) correct, ignoring some important literature in federated learning.
  - In lines 185-187, the authors mentioned, "However, existing methods for personalization in conventional FL often fall short in terms of generalization (Jiang & Lin, 2023; Xie et al., 2024), making them less effective for the versatile applications required in FedFM." However, there are some important works that discuss the relationship between generalization and personalization, and in some sense, both can be improved simultaneously by one method. You can refer to FedRoD [2] and FedETF [3].
  - The authors mentioned invariant learning in federated learning as their motivation. However, the most important literature is missing here. The authors should refer to DFL [4], cite it, discuss it, and compare it.
- **Technical novelty**. The main methodology of this paper is to use the global prototype as a regularization for local features. However, similar ideas already existed [5]. Therefore, the technical novelty is marginal here.

---
[1] Ye, Rui, et al. "Openfedllm: Training large language models on decentralized private data via federated learning." Proceedings of the 30th ACM SIGKDD Conference on Knowledge Discovery and Data Mining. 2024.

[2] Chen, Hong-You, and Wei-Lun Chao. "On bridging generic and personalized federated learning for image classification." ICLR 2022.

[3] Li, Zexi, et al. "No fear of classifier biases: Neural collapse inspired federated learning with synthetic and fixed classifier." Proceedings of the IEEE/CVF International Conference on Computer Vision. 2023.

[4] Luo, Zhengquan, et al. "Disentangled Federated Learning for Tackling Attributes Skew via Invariant Aggregation and Diversity Transferring." International Conference on Machine Learning. PMLR, 2022.

[5] Zhou, Tailin, Jun Zhang, and Danny HK Tsang. "FedFA: Federated learning with feature anchors to align features and classifiers for heterogeneous data." IEEE Transactions on Mobile Computing (2023).

**Questions:**

See weaknesses.

---

> ### Author Response · Authors · 2024-11-23
>
> Dear Reviewer DApR,
>
> Thanks for your valuable reviews, however, there are some misunderstandings about our paper and detailed responses to each question are provided below.
>
> **W1: Theoretical analysis is not related to the foundation model properties.**
>
> >**Our Theorem 1 and Theorem 2 are based on foundation model properties**, which mainly rely on the pretrained head of foundation models promoting only the feature encoder to handle out-of-distribution (OOD) by learning invariant features. Specifically,  the pretrained head of foundation models is fixed and taken as the optimal head for all tasks to reduce prediction disagreement for implicitly invariant features learning, similar inspiration as previous work [1]. Additionally, with the pretrained head, the generalization bound of the personalized model in FedFM can be further constrained by the invariant feature distance as illustrated in Theorem 2. However, conventional FL does not have the pretrained knowledge to guarantee invariant learning. We add more explanation of Theorem 1 and 2 in Section 3.1 and Appendix D for better understanding.
>
> **W2: Do not compare with conventional FL baselines.**
>
> > **We do compare our method with conventional FL methods adapted to FedFM**, as described in **Section 5.1** and **Appendix C.2**. The conventional FL methods we compared—pFedMe[2], FedLoRA [3], PRADA [4], and FedSDR [5]—have been adapted to FedFM with adapters for a fair comparison. The results shown in Table 2 support our claim that conventional FL methods are suboptimal for addressing OOD generalization in FedFM.
>
> **W3: The inappropriate use of foundation models.**
>
> >We are focusing on the research direction termed as federated foundation models (FedFM) [6], where most foundation models are predominantly based on transformer architectures with large pretrained data originating from NLP. **Our theories and methods are all universally applicable towards FedFM and do not rely on any specific characteristics of a particular subdomain.** Similar to previous conventional FL methods (e.g. SCAFFOLD [7]) only conducting experiments in a subdomain (image models) for FL, we also utilize model from a subdomain (LLM from NLP) as exemplary instances for FedFM to validate our proposed theories and methods, and LLaMA stands out as one of the most popular open-source foundation models, boasting significantly greater parameter volumes (hundreds to thousands of times larger) compared to models like ViT or CLIP.
>
> >In addition, our method can be easily adapted to other transformer-based foundation models by simply replacing the LLaMA with other models like ViT. We also conduct experiments with ViT on OfficeHome dataset [8] as shown in the following table. The results show our method could outperform other personalized models (pFedMe, FedLoRA and PRADA) and have comparable results with global model (FedIT), which demonstrates the effectiveness of our methods towards different foundation models. Specifically, our proposed invariant feature-based regularization is designed to enhance the personalized model's OOD generalization ability by leveraging the global model's generalization capabilities. Since we employ FedIT as the global model, we expect that our personalized models will exhibit OOD generalization results comparable to those of FedIT. We have added a discussion in Appendix F.2 to discuss this.
> >
> >| Methods    | Art | CliPart | Product | Real World | Avg |
> >| -------- | ------- | -------- | ------- | -------- | ------- |
> >| FedIT  |  68.11  | 56.66 | 77.18 | 77.94 | 69.97 |
> >| pFedMe | 54.72  | 41.25 | 59.22 | 60.67 | 53.96 |
> >| FedLoRA   | 60.49 | 51.31 | 72.93 | 73.15 | 64.47 |
> >| PRADA | 54.73 | 41.25 | 59.24 | 60.68 | 53.96 |
> >| FedOA | 67.49 | 56.51 | 75.96 | 77.45 | **69.35**|

---

> > ### Author Response · Authors · 2024-11-23
> >
> > **W4: Lack of important literature (2-5).**
> >
> > >Our paper mainly focuses on **out-of-distribution (OOD) generalization** following previous work [1,5], while all the literature you list (2-5) focuses on **in-distribution performance**, which are two different research directions. In FL, OOD generalization focuses on testing on new environments unseen for all training clients, while in-distribution focuses on testing on seen environments. The term “generalization” in our paper refers to OOD generalization, while “generalization” in the literature you list refers to the in-distribution performance of personalized/global model testing data from other/all training clients’ seen environments. For example, consider an image classification system with two clients obtaining data from different domains (Art and Clipart), OOD generalization is to deal with new domains like images from Real World, while in-distribution generalization assesses the performance of the global model for both Art and Clipart domains or the Art client tested on Clipart domain. More importantly, we also include the most recent benchmark PERADA [4] of in-distribution generalization for comparison in Table 2, which shows poorer performance than our method to support our claim. We added more discussions in our related work in Appendix B.2 to clarify.
> >
> > **W5: Technical novelty compared with previous work (5) using global prototype as regularization.**
> >
> > >- Our research specifically targets OOD generalization, which contrasts with the paper you listed that concentrates on in-distribution performance.
> > >
> > >- **We do not use prototypes but invariant features as regularization.** Prototypes necessitate a finite categorization and are unsuitable for foundation models in OOD scenarios due to open-vocabulary tasks inherently (e.g. the categories of real-world images are effectively infinite). Unlike prototypes that demand a specific prototype for each category, invariant features are not bound by a set number of categories and are learned autonomously across different environments by the feature encoder. We add this explanation in Section 3.2 for clarity.
> > ---
> > [1] Guo, Yaming, et al. "Out-of-distribution generalization of federated learning via implicit invariant relationships." International Conference on Machine Learning. PMLR, 2023.
> >
> > [2] T Dinh, Canh, Nguyen Tran, and Josh Nguyen. "Personalized federated learning with moreau envelopes." Advances in neural information processing systems 33 (2020): 21394-21405.
> >
> > [3] Yi, Liping, et al. "Fedlora: Model-heterogeneous personalized federated learning with lora tuning." arXiv preprint arXiv:2310.13283 (2023).
> >
> > [4] Xie, Chulin, et al. "PerAda: Parameter-Efficient Federated Learning Personalization with Generalization Guarantees." Proceedings of the IEEE/CVF Conference on Computer Vision and Pattern Recognition. 2024.
> >
> > [5] Tang, Xueyang, et al. "Learning personalized causally invariant representations for heterogeneous federated clients." The Twelfth International Conference on Learning Representations. 2023.
> >
> > [6] Ren, Chao, et al. "Advances and open challenges in federated learning with foundation models." arXiv preprint arXiv:2404.15381 (2024).
> >
> > [7] Karimireddy, Sai Praneeth, et al. "Scaffold: Stochastic controlled averaging for federated learning." International conference on machine learning. PMLR, 2020.
> >
> > [8] Venkateswara, Hemanth, et al. "Deep hashing network for unsupervised domain adaptation." Proceedings of the IEEE conference on computer vision and pattern recognition. 2017.

---

> > > ### Author Response · Authors · 2024-11-26
> > >
> > > Dear Reviewer DApR,
> > >
> > > We genuinely hope that our response can address your concerns. If you have further questions, we can discuss them in this phase.

---

> > > > ### Author Response · Authors · 2024-12-01
> > > >
> > > > Dear Reviewer DApR,
> > > >
> > > > As the deadline for the Reviewer-Author discussion phase is fast approaching (there is only a day left), we respectfully ask whether we have addressed your questions and concerns adequately.

---

### Official Review · Reviewer_DPmc · 2024-11-01

**Soundness:** 3
**Presentation:** 3
**Contribution:** 3
**Rating:** 6
**Confidence:** 4

**Summary:**

The paper addresses the challenge of out-of-distribution (OOD) generalization in Federated Foundation Models (FedFM), which are affected by distribution shifts, large parameter scales, and data heterogeneity, leading to suboptimal client performance. To tackle these issues, the authors propose FedOA, an invariant learning-based approach that employs adapter-based fine-tuning for efficiency and incorporates a personalized model with feature distance-based regularization to improve OOD generalization across clients. The paper provides theoretical guarantees on OOD generalization and convergence in non-convex settings, and empirically demonstrates that FedOA outperforms existing methods on benchmark NLP tasks.

**Strengths:**

(a) The paper is well-written with a clear structure and logical flow, making it easy to understand the key contributions, motivation, and main theoretical and experimental findings. Additionally, the appendix provides well-organized details on the experimental setup and proofs of theorems.

(b) This work includes a rigorous theoretical convergence analysis for the proposed method and conducts evaluations on advanced NLP federated learning tasks using large language models.

(c) The ablation study is comprehensive, covering key hyper-parameters as well as detailed convergence and generalization analyses.

**Weaknesses:**

(a) The connection between the proposed invariance-based FL regularization and its motivation could be clarified. It is not evident how the approach specifically addresses the challenge of large parameter scales in federated learning with foundation models, as the parameter scale issue seems to be tackled largely through the integration of parameter-efficient fine-tuning rather than through the regularization method itself.

(b) The evaluation dataset is limited in size and scope, covering only a few domains, which may not sufficiently showcase the method’s effectiveness in more diverse or large-scale federated learning environments.

**Questions:**

See weakness

---

> ### Author Response · Authors · 2024-11-23
>
> Dear Reviewer DPmc,
>
> Thanks for your valuable reviews and  your recognition of the key contributions of our work. Detailed responses to each question are provided below.
>
> **W1: The motivation of the proposed invariance-based regularization.**
>
> >We propose a new framework towards OOD generalization of FedFM, both the integration of PEFT methods and the proposed invariance-based regularization are important parts of our design of the proposed framework. Yes, the integration of PEFT methods primarily addresses issues related to parameter scale, while the proposed invariance-based regularization, based on PEFT methods, targets OOD generalization issues. Furthermore, as explained in “Why feature distance-based regularization? ” of Section 3.2, the proposed invariance-based regularization could further address large parameter scale issues by using  features for calculation, which are significantly smaller in size than parameters for computations in federated foundation models. Thanks for your advice, we have revised the introduction to make our motivation more clear.
>
> **W2: Evaluated on large-scale FL.**
>
> >Yes, evaluation on a large-scale FL could further demonstrate our method’s effectiveness. However, given the constraints of computational resources and the use of large foundation models with billions of parameters, our experiments are strategically designed to validate the efficacy of our method and the soundness of the proposed theorems with limited resources. Additionally, we have conducted supplementary experiments that expand the number of clients to 30, as detailed in the table below, and compared the results with other baseline methods on reading comprehension (RC) dataset. The findings from these experiments underscore the scalability of our method. We have added this discussion in Appendix F.1.
> >
> >|    | FedIT | PRADA | FedLoRA | FedOA |
> >| -------- | ------- | -------- | ------- | -------- |
> >| RC  | 58.04 | 39.30 | 46.90 | **58.84** |

---

> > ### Comment · Reviewer_DPmc · 2024-11-26
> >
> > Thank you for addressing my concerns. I am satisfied with your response, and the issue mentioned above has been resolved. I will maintain my positive score for acceptance.

---

### Official Review · Reviewer_Um8N · 2024-11-04

**Soundness:** 2
**Presentation:** 2
**Contribution:** 2
**Rating:** 5
**Confidence:** 4

**Summary:**

This paper investigates the personalization and out-of-distribution generalization of adapter-based foundation models during federated fine-tuning. To tackle the data heterogeneity across local clients, the proposed framework targets at develops a personalized (fine-tuned) model for each local client, with the guidance of a global fine-tuned model. The global model is fine-tuned using the conventional federated learning scheme (i.e., minimizing the weighted empirical loss), while the personalized models are fine-tuned with a constraint that penalizes the distance between global representations and personalized representations. A theoretical analysis of the convergence rate of the proposed algorithm FedOA is provided, and its empirical performance is evaluated on four NLP datasets.

**Strengths:**

1. The proposed algorithm is straightforward and can be easily integrated with existing federated learning methods.

2. In designing the distance-based regularization term for training personalized models, the structural heterogeneity across the parameter-efficient fine-tuning (PEFT) methods used by local clients is taken into account.

3. This paper provides a theoretical guarantee on the convergence rate of the proposed algorithm.

**Weaknesses:**

1. The novelty and contributions of this paper are limited. The proposed method, FedOA, is based on the existing federated foundation model scheme, FedIT [1], with the primary distinction being the introduction of a distance-based regularization term for training personalized models.

2. The conclusion in Theorem 1 follows straightforwardly from the theoretical results presented in prior work [2]. Furthermore, minimizing the weighted empirical loss (i.e., objective (2) on page 3) does not ensure that the global model captures invariant representations. Consequently, the generalization performance of the proposed method cannot be guaranteed.

3. Since minimizing objective (2) on page 3 does not ensure that the extracted features satisfy Assumption 1, the conclusions in both Theorem 1 and Theorem 2 do not hold for the proposed algorithm. Regarding how to ensure that the invariance constraint in Assumption 1 is satisfied, it is recommended to refer to the literature on invariant learning, such as [3][4], for further details.

4. Additional details on the structure of the adopted PEFT framework should be included in the main text to aid understanding.

5. In the evaluation section, performance under partial client participation and scalability with a large number of clients is not assessed, which is significant for the applicability of the proposed algorithm in practical federated learning scenarios.

6. It appears that the setup of the test dataset for each client is not discussed in the experimental section.


[1] Zhang, J., Vahidian, S., Kuo, M., Li, C., Zhang, R., Yu, T., ... & Chen, Y. (2024, April). Towards building the federatedGPT: Federated instruction tuning. In ICASSP 2024-2024 IEEE International Conference on Acoustics, Speech and Signal Processing (ICASSP) (pp. 6915-6919). IEEE.

[2] Nikola Konstantinov and Christoph Lampert. Robust learning from untrusted sources. In Interna- tional conference on machine learning, pp. 3488–3498. PMLR, 2019.

[3] Arjovsky, M., Bottou, L., Gulrajani, I., & Lopez-Paz, D. (2019). Invariant risk minimization. arXiv preprint arXiv:1907.02893.

[4] Rosenfeld, E., Ravikumar, P., & Risteski, A. (2020). The risks of invariant risk minimization. arXiv preprint arXiv:2010.05761.

**Questions:**

Please refer to weaknesses 1-6 outlined in the previous section.

---

> ### Author Response · Authors · 2024-11-23
>
> Dear Reviewer Um8N,
>
> Thanks for your valuable reviews, detailed responses to each question are provided below.
>
> **W1: Limit novelty and contributions.**
>
> >Our paper is to further explore federated foundation models by proposing a new framework to consider out-of-distribution (OOD) generalization with theoretical guarantees.
> >
> > - Our paper mainly focuses on OOD generalization of federated foundation models, while FedIT focuses on in-distribution performance.
> > - Our paper considers OOD generalization scenarios for both conventional and personalized FL, while FedIT only addresses conventional FL.
> >- We contribute theoretical analysis covering both OOD generalization and convergence, whereas FedIT lacks such theoretical analysis.
> >- We introduce a novel bi-level optimization framework targeting both global and personalized adapters for OOD generalization, in contrast to FedIT's singular focus on an aggregated objective for global adapter optimization on in-distribution performance.
>
> **W2:  Objective (2) cannot learn invariant features.**
>
> >**Objective (2) focuses solely on the empirical loss related to the feature encoder** $\Phi$, rather than on the empirical loss associated with the entire model $f$. This distinction implies that **the empirical loss labels for the feature encoder of Objective (2) are based on features $z$ that are consistent across different environments** (invariant features), rather than on the overall model empirical loss labels $y$ (e.g., [0,1]).
>
> >Objective (2) provides a succinct summary of previous FL work that employed invariant learning approaches (**not only Invariant risk minimization**). For instance, previous study [1] achieves objective (2) by directly using contrastive learning to narrow the gap between features of the same category across different domains and to distance features of unrelated categories, thus facilitating the learning of invariant features. In addition, work [2] implicitly achieves objective (2) by aligning inter-client’s heads to reduce prediction disagreement for implicitly invariant features learning. We add more explanation of the objective (2) in Section 2.1 for better understanding.
>
> **W3: Theorem 1 directly from previous work and Theorem 1, 2 does not hold.**
>
> >Theorem 1 seeks to extend the theoretical results of previous work to federated foundation models settings. In Theorem 1, we claim that through the aggregation of federated foundation models with a pretrained head, it is possible to directly fulfill Objective (2). That is, during tuning, **the pretrained head of foundation models is fixed and taken as the optimal head for all tasks** ($w \in argmin_{w} R_e(w,\Phi_g)$ for all $e \in \mathcal{E}_{all}$) to implicitly learn invariant features, directly achieving the objective (following equation) like invariant risk minimization. By omitting the pretrained head, the objective of the aggregation of federated foundation models can be simplified as objective (2). This approach draws similar inspiration from previous work [2].
>
> >$\min_{\Phi_g} \sum_{e\in \mathcal{E}_{train}}\alpha_e R_e(w,\Phi_g)$
> >
> >s.t. $w \in argmin_{w} R_e(w,\Phi_g)$ for all $e \in \mathcal{E}_{train}$
>
> >Based on the theoretical results of previous work, the generalization bound of the global model in FedFM can be further bounded as $Supp_{\cup\Phi}(|R_e(\Phi)-R_{e'}(\Phi)|)$. For easy understanding, we add more explanation of Theorem 1 in Section 3.1 and Appendix D.
>
> **W4: Details of the adopted PEFT framework.**
>
> >Our methods are designed with high flexibility, allowing for the integration of various adapter-based PEFT methods, and all these adapter-based PEFT methods are unified under a common scheme that partitions parameters into two parts: one small trainable part and one large frozen part. Thanks for your advice, we already add more details of the adopted PEFT framework in Section 3.2.
>
> **W5: Scalability with larger clients.**
>
> >Given the constraints of computational resources associated with large foundation models containing billions of parameters, our experiments are mainly designed in an ideal setting to validate the efficacy of our method and the proposed theorems. In addition, to demonstrate our method’s scalability, we conduct additional experiments involving an expanded group of 30 clients as detailed in the following table, compared with other baselines on reading comprehension task (RC). The results from this table further demonstrate the effectiveness of our method. We have added this discussion in Appendix F.1.
> >
> >|    | FedIT | PRADA | FedLoRA | FedOA
> >|  ----  | ----  | ----  | ----  | ----  |
> >| RC | 58.04| 39.30 | 46.90 | **58.84**|

---

> > ### Author Response · Authors · 2024-11-23
> >
> > **W6: Setup of test data.**
> >
> > >Our setup of the test dataset is detailed in Experiment Setting of **Section 5.1** and **Appendix C.1**. We constructed four tasks (each task contains two datasets with split training and testing data for each dataset) from Flan as presented in Table 5. We used a “leave-one-task-out” strategy, where one task is set aside as the test environment, while the remaining are used as training environments. For example,  if the task of Entailment (comprising test instances from the snli and anli datasets) is selected as the test dataset, then the remaining six datasets of three tasks (Sentiment, Paraphrase and Reading Comprehension) are used for training with each client contains one dataset. We add this example for better understanding in Appendix C.1.
> > ---
> > [1] Tan, Yue, et al. "Is heterogeneity notorious? taming heterogeneity to handle test-time shift in federated learning." Advances in Neural Information Processing Systems 36 (2024).
> >
> > [2] Guo, Yaming, et al. "Out-of-distribution generalization of federated learning via implicit invariant relationships." International Conference on Machine Learning. PMLR, 2023.

---

> > > ### Author Response · Authors · 2024-11-26
> > >
> > > Dear Reviewer Um8N,
> > >
> > > We genuinely hope that our response can address your concerns. If you have further questions, we can discuss them in this phase.

---

> > > > ### Comment · Reviewer_Um8N · 2024-11-30
> > > >
> > > > Thanks for the authors' detailed responses. The provided explanations and additional experiments have addressed most of my concerns. After careful consideration, I'm willing to update the score and increase my rating to 5.

---

> > > > > ### Author Response · Authors · 2024-12-01
> > > > >
> > > > > Dear Reviewer Um8N,
> > > > >
> > > > > Thank you for acknowledging our efforts and the improvements made to our paper. We appreciate your feedback and kindly ask if there are any remaining major issues preventing our paper from being considered acceptable. We are open to discussing any remaining concerns you might have and would greatly value any additional feedback to enhance our work. If all concerns are adequately addressed, we would greatly appreciate it if you could consider our paper for acceptance.

---

### Comment · Area_Chair_RZhi · 2024-12-01

Dear Reviewers,

This is a reminder that December 2 is the final day to post feedback to the authors.

Your input is critical to the review process. If you have not yet responded, we kindly encourage you to confirm whether you have reviewed the rebuttals. Additionally, if you have any remaining concerns or clarifications, please share them with the authors before the deadline.

Thank you for your valuable contributions.

Best,

Your AC

---

### Meta-Review · Area_Chair_RZhi · 2024-12-21

**Metareview:**

This paper investigates OOD generalization in personalized FedFM through the introduction of FedOA, a framework that combines adapter-based fine-tuning with feature distance-based regularization to mitigate client data heterogeneity. The reviewers appreciated the paper's clear presentation and acknowledged the effort to address identified weaknesses. However, the revisions fall short of the standards expected for ICLR publication. The work would benefit from further refinement by providing clearer justification of its theoretical and methodological novelty in relation to existing approaches, offering stronger motivation for the problem setting and proposed methods, conducting more comprehensive evaluations, and thoroughly discussing relevant literature.

**Additional Comments On Reviewer Discussion:**

During the discussion phase, Reviewer Um8N acknowledged that some of their concerns were addressed and adjusted their score to 5. Reviewer DPmc maintained their score of 6 but the weakness pointed out were not very critical. While Reviewer DApR did not respond, after carefully evaluating the authors' responses, I remain unconvinced by their justification of the theoretical analysis and their rationale for excluding related literature, particularly given the overlap in techniques with prior work. These gaps weaken the paper’s claim of novelty and its alignment with the standards of the venue.

---

### Decision · Program_Chairs · 2025-01-22

Reject